Bodily action penetrates affective perception

Fantoni Carlo cfantoni@units.it
Rigutti Sara
Gerbino Walter
Department of Life Sciences, Psychology Unit “Gaetano Kanizsa,” University of Trieste , Trieste , Italy
Borghi Anna
Electronic publication date: 2016 Feb 15
Publication date: 2016
Volume: 4
Electronic Location ID: e1677
Received 2015 Oct 30; Accepted 2016 Jan 20
Copyright: © 2016 Fantoni et al.
Copyright year: 2016
Copyright holder: Fantoni et al.
License: This is an open access article distributed under the terms of the Creative Commons Attribution License, which permits unrestricted use, distribution, reproduction and adaptation in any medium and for any purpose provided that it is properly attributed. For attribution, the original author(s), title, publication source (PeerJ) and either DOI or URL of the article must be cited.
License URL: https://creativecommons.org/licenses/by/4.0/

Keywords: Mood, Face perception, Perception, Penetrability, Emotion, Action, Induction, Motor, Adaptation, Embodied

Funding: University of Trieste FRA-2013 This work was supported by the University of Trieste (FRA-2013 Grant to CF). The funders had no role in study design, data collection and analysis, decision to publish, or preparation of the manuscript.

==============================
Fantoni & Gerbino (2014) showed that subtle postural shifts associated with reaching can have a strong hedonic impact and affect how actors experience facial expressions of emotion. Using a novel Motor Action Mood Induction Procedure (MAMIP), they found consistent congruency effects in participants who performed a facial emotion identification task after a sequence of visually-guided reaches: a face perceived as neutral in a baseline condition appeared slightly happy after comfortable actions and slightly angry after uncomfortable actions. However, skeptics about the penetrability of perception (Zeimbekis & Raftopoulos, 2015) would consider such evidence insufficient to demonstrate that observer’s internal states induced by action comfort/discomfort affect perception in a top-down fashion. The action-modulated mood might have produced a back-end memory effect capable of affecting post-perceptual and decision processing, but not front-end perception.

Here, we present evidence that performing a facial emotion detection (not identification) task after MAMIP exhibits systematic mood-congruent sensitivity changes, rather than response bias changes attributable to cognitive set shifts; i.e., we show that observer’s internal states induced by bodily action can modulate affective perception. The detection threshold for happiness was lower after fifty comfortable than uncomfortable reaches; while the detection threshold for anger was lower after fifty uncomfortable than comfortable reaches. Action valence induced an overall sensitivity improvement in detecting subtle variations of congruent facial expressions (happiness after positive comfortable actions, anger after negative uncomfortable actions), in the absence of significant response bias shifts. Notably, both comfortable and uncomfortable reaches impact sensitivity in an approximately symmetric way relative to a baseline inaction condition. All of these constitute compelling evidence of a genuine top-down effect on perception: specifically, facial expressions of emotion are penetrable by action-induced mood. Affective priming by action valence is a candidate mechanism for the influence of observer’s internal states on properties experienced as phenomenally objective and yet loaded with meaning.

Introduction

Penetrability of perception (Firestone & Scholl, 2015; Gerbino & Fantoni, 2016; Zeimbekis & Raftopoulos, 2015) refers–among others–to the possible effects of bodily actions on affective perception (i.e., on aspects of perceived objects related to valence and arousal). Compared to the link between bodily actions (or intentions to activate the body; Vishton et al., 2007; Briscoe, 2014) and perception of spatial/material properties (Witt, 2011; Zadra & Clore, 2011), the link between bodily actions and affective perception appears quite plausible, being grounded in the phenomenal experience of bodily comfort/discomfort associated to motor actions. The emotional coloring of perception by observer’s bodily actions could fall in the generic category of halo effects or attribution errors (consisting in, for instance, inappropriately attributing to external objects what should more reasonably pertain to the observing self).

Reading emotion in others’ faces is a relevant instance of affective perception. However, despite the pervasive role of action in object representation (Santos & Hood, 2009), little is known about the link between action and perception of facial expressions. This link is crucial in ontogenesis, as caregivers’ responses to infant actions are the building blocks of socio-emotional development, visuo-motor control, and structuring of external object properties (Farroni et al., 2002; Leppänen & Nelson, 2009; Craighero et al., 2011; Grossmann et al., 2013). As a case in point, the categorical perception and representation of emotionally expressive faces have been found to depend on bodily affective experience: children exposed to negative affective experience (e.g., physical abuse) over-identified anger relative to control children and produced discrimination peaks reflecting broader perceptual categorization of anger, relative to fear and sadness (Pollak & Kistler, 2002). This is consistent with the James-Lange theory of emotion (James, 1884) and recent evidence supporting the view that emotions are represented as culturally universal somatotopic maps; i.e., topographically distinct bodily sensations underlying the categorical perception of different emotions (Nummenmaa et al., 2014).

Bodily actions are both informative (about object properties) and affective, given that the experience of comfort/discomfort is a pervasive component of body feelings associated to action execution (Mark et al., 1997), which can propagate from directly involved effectors to the whole body and transient mood (Cahour, 2008). In the workplace a posture that imposes musculoskeletal discomfort has been found to induce a negative mood capable of impairing psychological health (Conway, 1999). Body posture affects behavior through transient mood in other ways: for instance, by modifying observers’ judgment of neutral objects like ideographs (Cacioppo, Priester & Berntson, 1993), as well as the subjective attitude towards dishonesty (Yap et al., 2013). Actions also manifest their effects on behavior implicitly, as revealed by the different parameterization of hand movement kinematics in presence/absence of a social context (Becchio et al., 2008a; Becchio et al., 2008b; Sartori et al., 2009; Ferri et al., 2011; Quesque et al., 2013; Straulino, Scaravilli & Castiello, 2015), end-goal accuracy (Ansuini et al., 2006), and motor affordances (Masson, Bub & Breuer, 2011; Tucker & Ellis, 1998; Tipper, Howard & Jackson, 1997; Glover et al., 2004) determined on the basis of biomechanical compatibility, relative to size, shape, and material properties of the object-hand system (Mon-Williams & Bingham, 2011; Flatters et al., 2012; Holt et al., 2013).

As regards the environment on which actions are directed to, while affective micro-valences are found in various types of everyday objects (Lebrecht et al., 2012), faces of conspecifics are among the stimuli with the most extreme affective valences. Looking at emotional faces has indeed been shown to potentiate the sensory discrimination benefits of attention by improving dimensions of early vision such as contrast sensitivity (Phelps, Ling & Carrasco, 2006). This is consistent with recent results suggesting that some components of facial expressions of different emotions (e.g., widened/narrowed eyes in fear/disgust) involve different exposures of iris and sclera that influence peripheral target discrimination by the expresser as well as the amount of optical information available to the observer (Lee, Susskind & Anderson, 2013).

Recently, Fantoni & Gerbino (2014) demonstrated that the internal state of comfort/discomfort induced by reaching affects the identification of facial expressions in a direction congruent with transient mood (Fantoni, Cavallero & Gerbino, 2014a; Gerbino et al., 2014a; Gerbino et al., 2014b). To manipulate transient mood in a controlled exploratory-action setting, Fantoni & Gerbino (2014) and implemented the Motor Action Mood Induction Procedure (MAMIP). The procedure required participants to perform a series of either comfortable or uncomfortable visually-guided reaches of targets randomly located at short vs. long distances, corresponding to [0.65–0.75] vs. [0.9–1.00] ranges, relative to arm length. Such a manipulation of distance was based on previous studies directly relating the subjective state of comfort/discomfort to the individual reaching mode, with perceived discomfort increasing as the number of body parts (muscles, joints) engaged in reaching increases (Mark et al., 1997).

The effectiveness of MAMIP was tested using a facial emotion identification task. After having performed a sequence of mood-inducing actions (either comfortable or uncomfortable), participants classified morphed faces displaying mixed expressions along the happy-to-angry continuum as either “happy” or “angry.” Systematic shifts of the point of subjective neutrality between these facial expressions were fully consistent with a facilitation-by-congruency hypothesis: comfortable actions increased the probability of classifying a neutral face as happy (positive emotion); whereas uncomfortable actions increased the probability of classifying a neutral face as angry (negative emotion). Furthermore, Fantoni & Gerbino (2014) found that JNDs were smaller when facial emotion identification was preceded by reaching, relative to a baseline inaction condition. They argued that hyperarousal from action (relative to inaction) improved the sensitivity to subtle variations of facial expression and reduced the degree of classification uncertainty (response times were shorter after reaching than in the baseline inaction condition). Finally, facial emotion identification was more precise (lower JND) and faster after uncomfortable than comfortable reaches, consistently with higher activation/arousal after uncomfortable motor actions.

This effects of motor action valence (in particular, comfort/discomfort of visually-guided reaches) on perceived facial expressions fit well in the framework of the emotional mirror system (Bastiaansen, Thioux & Keysers, 2009) and are consistent with emotional response categorization theory (Niedenthal, Halberstadt & Innes-Ker, 1999), which implies that humans are tuned to respond to conspecifics in a way that is congruent with their emotional state. Specifically, Fantoni & Gerbino (2014) suggested that action comfort/discomfort may indirectly prime the successive identification of emotions through transient mood induction (for a review of direct and indirect affective priming effects see Janiszewski & Wyer, 2014).

Expressive faces have indeed been used to demonstrate the occurrence of mood-dependent face categorization using music as an inducer of internal states (Bouhuys, Bloem & Groothuis, 1995). Furthermore, the subjective state of comfort/discomfort has been found to correlate with mood (Conway, 1999) and to depend upon the individual reaching mode, with discomfort being a direct function of the amount of body movement supplementary to arm extension (Mark et al., 1997). The relationship between reaching distance and induced state of comfort/discomfort is mirrored by the way the planning of hand movements with different movement amplitudes (involving variable amounts of motor effort, muscular strength, and number of involved joints in the act) has been recently shown to affect perception (Kirsch & Kunde, 2013; Volcic et al., 2013). Specifically, Kirsch & Kunde (2013) have shown that larger planning of goal-directed hand movements obtained by instructing observers to perform movements with larger extent causes the same spatial location to appear as further away. Similarly, Volcic et al. (2013) found that the repeated execution of reaching movements with the visual feedback of the reaching finger displaced farther in depth as if the actor would have a longer arm, enhanced both tactile sensitivity and the perceived depth of disparity-defined objects.

However, no clear evidence allows us to decide whether changes in identification performance obtained by Fantoni & Gerbino (2014) depend on stimulus encoding, as involving a modification of the perceptual processing of facial emotion features, or response selection, as involving only a post-perceptual modification of the response criterion and decision thresholds (Spruyt et al., 2002).

Are MAMIP effects truly perceptual?

The effectiveness of MAMIP has been tested by Fantoni & Gerbino (2014) under the assumption that the repeated execution of motor actions with a variable perceived degree of comfort generates a transient mood shift consistent with affective adaptation (Wilson & Gilbert, 2008); i.e., with the progressive reduction of action valence paralleling a prolonged induction. Affective adaptation is a convenient umbrella expression, including anecdotal evidence that the valence of objects and events tends to fade away (i.e., tends to become less extreme) during prolonged exposure, as a possible consequence of a shift of the internal reference level involved in the evaluation of perceived objects and events. In the domain of motor actions, the repetition of comfortable reaches should: (i) induce a better mood in the actor; and (ii) make comfortable reaches progressively closer to neutrality (i.e., less discrepant from the shifted internal reference level). Likewise, the repetition of uncomfortable reaches should: (i) induce a worse mood in the actor; and (ii) make uncomfortable reaches progressively closer to neutrality (i.e., less discrepant from the shifted internal reference level).

Transient mood shifts modify the valence of not only bodily actions but also external objects; notably, expressive faces. Mood-congruent perception has been frequently reported in the literature (Bouhuys, Bloem & Groothuis, 1995). Such an effect is consistent with the notion of affective priming (Klauer & Musch, 2003), given that the observer’s state induced by action valence can be treated as an affective prime, capable of pre-activating detectors selectively tuned to face features involved in the expression of congruent emotions. Consider a morphed face that interpolates the facial expressions of happiness and anger. Depending on observer’s mood, the interpolated face will contain features that support (in different proportions, by definition) both mood-congruent and mood-incongruent emotions: for instance, a cheek puffer should be more salient than an upper lip riser for an observer in a positive mood, while the opposite should hold for an observer in a negative mood.

According to a strong version of the facilitation-by-congruency hypothesis, the selective pre-activation of feature detectors by the transiently induced mood might produce two effects: an emotion identification bias (leading to a higher probability of classifying a neutral face as expressing a congruent emotion) as well as an improvement of the detection of a congruent emotion. According to a weak version of the same hypothesis, the transiently induced mood might produce only an emotion identification bias, without a change in sensitivity. This distinction is consistent with the categorization of priming effects proposed by Spruyt et al. (2002). According to the encoding stimulus account of affective priming, action valence acts as an affectively polarized prime that pre-activates the memory representations of affectively related facial features, thus making it easier to encode emotional features belonging to the same valence domain rather than to a different one. Alternatively, according to the response selection account of affective priming action affects only post-perceptual processing by automatically triggering response tendencies that facilitate or interfere with response types. Notice that in both cases one should expect a shift of the point of subjective neutrality due to an unbalance that depends on the valence congruency between the action-induced mood and the facial expression of emotion. Both accounts predict that the likelihood of interpreting a facial expression as angry, for instance, is larger after an uncomfortable rather than comfortable reaching act.

Fantoni & Gerbino (2014) demonstrated the effectiveness of MAMIP using a facial emotion identification (not detection) task. However, according to the above-described distinction and following the criticism of identification data voiced by Firestone & Scholl (2015), results obtained by Fantoni & Gerbino (2014) might simply reflect back-end memory processes rather than the penetrability of perception by internal states.

Furthermore, according to Fantoni & Gerbino (2014) the affective perception of faces is influenced by congruency between valence of performed actions and valence of the target facial expression (mediated by transient mood) as well as arousal. In principle, perceived facial expressions might be influenced by both mood valence (negative after uncomfortable acts vs. positive after comfortable acts) and arousal (higher after uncomfortable than comfortable acts). Independent of congruency between observer’s transient mood and emotion displayed by the target face, higher arousal might have facilitated processing of expressive faces evaluated after uncomfortable actions. The arousal-based hypothesis is consistent with evidence that visual sensitivity can be improved by hyperarousal induced by cold pressor stimulation (Woods et al., 2012; Woods, Philbeck, & Wirtz, 2013). In the experiment by Fantoni & Gerbino (2014), hyperarousal induced by uncomfortable actions might have influenced emotion identification by modulating selective attention (Derryberry & Reed, 1998; Gasper & Clore, 2002; Jeffries et al., 2008). Following Firestone & Scholl (2015), this interpretation would shed doubt on the truly perceptual origin of the effect found by Fantoni & Gerbino (2014), as modulations of selective attention could in principle change the input rather than subsequent processing.

To corroborate Fantoni & Gerbino’s (2014) results, in the present study we asked whether MAMIP effects reported in their paper are truly perceptual and whether evidence can be provided in favor of an encoding rather than response account of affective priming of perceived facial expressions of emotion, as mediated by action-induced mood states.

Method

Rationale and expectations

Experiments were designed to fulfil three goals: (G1) to disentangle the contributions of action valence and arousal to the perception of facial expressions of emotion; (G2) to decide whether action-induced mood congruency affects the valence of perceived expressions through a top-down modulation of perceptual processing, signalled by a modification of emotion sensitivity in the absence of response criterion shifts, or only affects a post-perceptual stage, by modifying the response criterion (i.e., a decision, not perception, threshold); (G3) to evaluate the contribution of action-induced mood congruency relative to the natural mood owned by a participant not performing any movement before the perceptual task.

To accomplish these goals we selected two emotions with opposite polarity in the valence domain (happiness and anger), an objective yes/no task, and a sensitivity experiment (according to the terminology suggested by Witt et al. (2015)) comprising positive (signal + noise, S + N) trials in which the face displayed a target emotion and negative (noise, N) trials in which the face displayed a neutral expression. Observers should be both fast and accurate when making their yes/no judgment on whether the face contained the emotion signal. Such a method minimized the role of cognitive top-down factors, unavoidable in the facial emotion identification procedure used by Fantoni & Gerbino (2014), and allowed us to decide whether MAMIP influenced emotion detection.

To measure sensitivity and bias in a facial emotion detection task we conducted four experiments with stimuli belonging to happiness and anger continua, generated by morphing a neutral face and an emotional face displaying either full happiness or full anger (Fig. 1). Four morphing levels were chosen for each continuum, corresponding to different per cent emotion: 0 (original neutral face), 10, 20, and 30% (Fig. 1B). Participants were tested individually in two successive sessions distinguished by reaches of opposite valence (comfortable/positive vs. uncomfortable/negative), with the ordering of action valences counterbalanced across participants. In doing so, we disentangled the independent contribution of mood congruency, action valence, and arousal, by using as between-subjects factors the temporal ordering of action type (comfortable ⇒ uncomfortable in Experiments 1 and 3 vs. uncomfortable ⇒ comfortable in Experiments 2 and 4) and the valence of the target emotion (positive in Experiments 1 & 2 vs. negative in Experiments 3 & 4). By combining action valence and target emotion valence in a crossover design we planned to test the effect of action/emotion valence congruency on the detection of both positive and negative emotions (Fig. 1A, cells of the two matrices). Specifically, we planned to contrast:– – in Experiments 1 & 2, happiness detection after congruent comfortable reaches (top-right cell of the left matrix & top-left cell of the right matrix) vs. incongruent uncomfortable reaches (top-left cell of the left matrix & top-right cell of the right matrix);

– – in Experiments 3 & 4, anger detection after congruent uncomfortable reaches (bottom-left cell of the left matrix & bottom-right cell of the right matrix) vs. incongruent comfortable reaches (bottom-right cell of the left matrix & bottom-left cell of the right matrix).

Figure 1 Experimental design and face set.

Panel (A) illustrates the mixed factorial design used in our study. The two temporal sequences in the top row (uncomfortable ⇒ comfortable on the left, comfortable ⇒ uncomfortable on the right) represent the levels of the between-subjects factor Action Ordering. Matrix rows represent the levels (happiness/positive on top, anger/negative on bottom) of between-subjects factor Target Emotion. Each cell of the two matrices encodes the two levels of the within-subjects factor Congruency, based on the correspondence between the valence of bodily motor acts performed before the emotion detection task (same along columns) and the valence of target emotion (same along rows). Panel (B) shows the faces of neutral-happy (top) and neutral-angry (bottom) morph continua for an exemplar character (not used in our experiments) who gave permission for the usage of his image (see Fantoni & Gerbino (2014), Fig. 1) for the characters of the Radboud database used in our experiments). The panel illustrates, for each continuum, the four morph intensities (0, 10, 20, 30%) used in the present study and the full emotion face (right), used to generate the blended items by morphing it with the neutral face (left).

We used an equal-variance signal detection model (Macmillan & Creelman, 2004) to obtain, for every participant, the individual values of three indices of performance extracted from the proportions of yes responses in emotion present trials (hits) and emotion absent trials (false alarms). Two indices refer to a perceptual stage: sensitivity d′ (one value for each of the three levels of emotion) and absolute threshold AT (defined as the per cent emotion value above which d′ gets positive). The third index, referring to the response stage, is criterion c.

After adaptation to comfortable/uncomfortable reaches under unrestrained body conditions, we measured detection performance at three increasing levels of emotion (10, 20, 30%) for two morph continua (positive vs. negative target emotion), to contrast two hypotheses about possible effects of motor action valence on sensitivity to facial expressions.

H1) According to the facilitation-by-congruency hypothesis (see Fig. 2), higher d′ and lower AT values were expected in congruent than incongruent conditions, independent of action valence per se, as follows: (1) performing comfortable rather than uncomfortable actions should improve happiness detection (Experiments 1 & 2); (2) vice versa, performing uncomfortable rather than comfortable actions should improve anger detection (Experiments 3 & 4). This hypothesis attributes the role of mediator to a transient modification of mood in a direction congruent with the performed action.

Figure 2 Alternative facilitation-by-congruency MAMIP effects.

Predictions resulting from additive (panels A, B) vs. multiplicative (panels D, E) MAMIP effects are shown in the sensitivity-by-morph spaces where d′ grows linearly as a function of per cent emotion in the morph for happy (panels A, D) and angry (panels B,E) faces. An additive MAMIP effect (A, B) is represented by parallel lines, with the red line (congruent/comfortable in (A) and congruent/uncomfortable in (B)) intersecting the per cent emotion axis at smaller values, than the cyan line (incongruent/uncomfortable in (A) and incongruent/comfortable in (B)): this produces a consistent shift in the absolute thresholds. A multiplicative MAMIP effect (D, E) is represented by lines diverging from a common origin lying along the per cent emotion axis that should produce constant absolute thresholds across different congruency conditions. The grey dotted line is the baseline performance if both comfortable and uncomfortable reaches impact sensitivity in a perfectly symmetric way. Predictions are recoded according to Equation 1 in the emotion detection space of panels (C, F) blue solid line stands for the line of comfortable reaches (happy congruent d′ as a function of angry incongruent d′); the violet solid line stands for the line of uncomfortable reaches (happy incongruent d′ as a function of angry congruent d′).

H2) According to the facilitation-by-arousal hypothesis, if uncomfortable reaches induce higher arousal in the observer and emotion detection depends on arousal (at least more than on transient mood), higher d′ and lower AT values were expected after uncomfortable actions, regardless of target emotion valence, as follows: performing uncomfortable rather than comfortable actions should improve detection of both happiness (Experiments 1 & 2) and anger (Experiments 3 & 4).

Note that the two hypotheses lead to the same expectations for anger detection (Experiments 3 & 4), but opposite expectations for happiness detection (Experiments 1 & 2). In the present study Action Ordering has been treated as a balancing between-subjects variable, while Target Emotion has been manipulated as a between-subjects factor to avoid a possible carryover effect intrinsic to an experimental design in which the same actors/observers detect emotions of opposite valence in successive sessions.

Furthermore, as depicted in Fig. 2 (panels A, D for happiness detection; panels B, E for anger detection), we considered two alternative possibilities about the MAMIP effect on AT, which was based on three d′ values:additive effect, producing different AT values in congruent (red solid line in Figs. 2A and 2B) and incongruent (cyan solid line in Figs. 2A and 2B) conditions, as a consequence of constant d′ increments/decrements at increasing per cent emotion in the morphed target; an additive effect on d′ would be signalled by significant main effects of morph intensity and of action/emotion valence congruency, in the absence of their interaction, in all four experiments;

multiplicative effect, producing the same AT value in congruent (red solid line in Figs. 2D and 2E) and incongruent (cyan solid line in Figs. 2D and 2E) conditions, as a consequence of proportionally increasing d′ increments/decrements as a function of per cent emotion in the morphed target; a multiplicative effect on d′ would be signalled by a significant morph intensity × action/emotion valence congruency interaction in all four experiments.

Note that both hypotheses hold if the assumption of d′ additivity is satisfied beyond MAMIP: The expected increase of sensitivity as a function of per cent emotion in the morph (as calculated by taking the proportion of false alarms in N trials as a common comparison value for the proportions of hits associated to each tested level of per cent emotion in S + N trials) should be approximately linear, with slope β2 and with negative intercept β1, thus allowing for the identification of an absolute threshold value above which d′ gets positive even in a baseline inaction condition (grey dotted line in Fig. 2).

Finally, we considered two alternative possibilities about the value of response criterion c, calculated by pooling together hits and false alarms for all three morph intensities, within each block of emotion detection trials following a MAMIP session. We expected that: (1) if MAMIP effects are perceptual (i.e., if bodily actions penetrate affective perception) then the c value should be constant across action/emotion valence congruency conditions associated with mood-congruent sensitivity effects; (2) if MAMIP effects are post-perceptual the c value should vary consistently with observer’s transient mood, even in the absence of an effect on sensitivity, due to a response bias in the direction of the action/mood congruent emotion (e.g., when required to detect anger observers might increase their yes rate after uncomfortable, relative to comfortable, actions).

Comparison with a baseline inaction condition

To accomplish G3 we compared the emotion sensitivity in Experiments 1–4 (after MAMIP) with that obtained in a preliminary experiment (baseline inaction condition, see Preliminary experiment section) on a large group of psychology undergraduates (N = 91, 66 females) who performed a 2AFC task not preceded by any motor task. Each 2AFC trial of the preliminary experiment included a neutral face and a morphed face with 15% emotion (anger or happiness). Participants should indicate which face (left/right) displayed an emotion.

We considered two possible outcomes of the comparison between Experiments 1–4 and the preliminary experiment, depending on whether the effect of action-induced mood on yes/no emotion sensitivity was symmetric or asymmetric, relative to the baseline inaction condition in which the mood was expected to be neutral (Figs. 2C and 2F). Consider comfortable reaches. If comfort empowers our sense of motor skillfulness, thus contributing to the establishment of a more positive mood than the one experienced in the baseline inaction condition, then in Experiments 1 & 2 happiness sensitivity after comfortable reaches (congruent condition) should be higher than happiness sensitivity in the preliminary experiment; conversely, in Experiments 3 & 4 anger sensitivity after comfortable reaches (incongruent condition) should be lower than anger sensitivity in the preliminary experiment. This hypothesis is based on the general idea that actions executed within the comfort range are rewarding and engagement in comfortable actions is more pleasant than comfort associated to inaction. Consider now reaching outside the natural grasping range, which should induce a negative mood as a direct product of discomfort or as an effect of experiential avoidance (Sloan, 2004). In Experiments 3 & 4 anger sensitivity after uncomfortable reaches (congruent condition) should be higher than anger sensitivity in the preliminary experiment, while in Experiments 1 & 2 happiness sensitivity after uncomfortable reaches (incongruent condition) should be lower than happiness sensitivity in the preliminary experiment.

A perfectly symmetric facilitation-by-congruency effect of action-induced mood on emotion sensitivity is illustrated in Fig. 2 according to additive (Figs. 2A and 2B) and multiplicative (Figs. 2D and 2E) models. The grey dotted line provides a reference along which the baseline d′ is expected to lie on. Considering a generic emotion detection Cartesian space with happiness d′ on the y axis and anger d′ on the x axis (Figs. 2C and 2F), this hypothesis leads to the following general expectation, irrespective on whether the MAMIP effect is additive (Fig. 2C) or multiplicative (Fig. 2F): the point representing performance in the baseline condition should lie in the portion of space between the blue solid line, depicting sensitivity after comfortable reaches (with anger sensitivity in the incongruent condition on the x axis and happiness sensitivity in the congruent condition on the y axis), and the violet solid line, depicting sensitivity after uncomfortable reaches (with anger sensitivity in the congruent condition on the x axis and happiness sensitivity in the incongruent condition on the y axis). These two lines can be recovered combining intercepts and slopes of the pair of lines best fitting the d′ triplets over the per cent emotion in the morph associated to comfortable (happy congruent and angry incongruent lines) and uncomfortable (happy incongruent and angry congruent lines) reaches. It can indeed be shown that slope β2 G and intercept β1 G of a line in the generic emotion detection space are given by:(1) β1G=β1happy−β2happyβ1angryβ2angryβ2G=β2happyβ2angry

where β1 happy and β2 angry correspond to the intercept and slope of the lines best fitting the average d′ triplets over morph intensity for happiness and anger sensitivity after mood-congruent and mood-incongruent reaches for the comfortable blue solid line, and vice versa for the uncomfortable violet solid line of Figs. 2C and 2F.

If the facilitation-by-congruency effect of action-induced mood on emotion sensitivity is perfectly symmetric, as exemplified in Fig. 2, then the baseline d′ should lie along the grey dotted line of either Fig. 2C, in the case of an additive effect, or Fig. 2F, in the case of a multiplicative effect, with the parameters of the grey dotted line resulting from averaging the parameters of the comfortable and uncomfortable lines. Otherwise, if the facilitation-by-congruency effect is asymmetric–possibly because comfortable reaches are equivalent to neutral inaction and do not modify the observer’s mood–then the baseline d′ should lie along the blue line of either Fig. 2C, in the case of an additive effect, or Fig. 2F, in the case of a multiplicative effect.

Participants

Forty undergraduates of the University of Trieste, all right handed, participated in the experiment lasting about one hour and half. All had normal or corrected-to-normal vision, were naïve to the purpose of the experiment, had no history of mood disorder and were not using antidepressant medications at the moment of the experiment. They were randomly assigned to one of the four experiments resulting from the combination of Action Ordering (comfortable first, uncomfortable first) and Target Emotion (happiness, anger). Each experiment included 10 participants distributed as follows across females and males: Experiment 1 (uncomfortable ⇒ comfortable, happiness; mean age = 23.7, SD = 2.5), females = 7; Experiment 2 (comfortable ⇒ uncomfortable, happiness; mean age = 24.3, SD = 2.1), females = 6, Experiment 3 (uncomfortable ⇒ comfortable, anger; mean age = 23.2, SD = 2.1), females = 6; Experiment 4 (comfortable ⇒ uncomfortable, anger; mean age = 23.5, SD = 1.5), females = 7.

The study was approved by the Research Ethics Committee of the University of Trieste (approval number 52) in compliance with national legislation, the Ethical Code of the Italian Association of Psychology, and the Code of Ethical Principles for Medical Research Involving Human Subjects of the World Medical Association (Declaration of Helsinki). Participants provided their written informed consent prior to inclusion in the study. The Ethics Committee of the University of Trieste approved the participation of regularly enrolled students to data collection sessions connected to this specific study, as well as the informed consent form that participants were required to sign. Dataset is available as a Supplemental data file (S1).

Apparatus, stimuli & design

The experimental setting utilized exactly the same Augmented Reality apparatus described in Fantoni & Gerbino (2014). Participants were seated in a dark laboratory in front of a high-quality, front-silvered 40 × 30 cm mirror, slanted at 45° relative to the participant’s sagittal body midline and reflecting images displayed on a Sony Trinitron Color Graphic Display GDM-F520 CRT monitor (19″; 1024 × 768 pixels; 85 Hz refresh rate), placed at the left of the mirror (Fantoni & Gerbino, 2014, Figs. 1B and 1C).

The same 3D visual displays used by Fantoni & Gerbino (2014) were used in the MAMIP reaching phases of the four experiments: (1) a high-contrast vertically oriented random-dot rod (30% density, visible back-surface, 7.5 mm radius, 65 mm height), depicted in Fig. 3A (see for details Fantoni & Gerbino, 2014, Fig. 1A); (2) a virtual red sphere (3 mm diameter) that visually marked the tip of the participant’s index finger in 3D space after the finger departed from its starting position of about 30 mm. Both stimuli were rendered in stereo and were generated using a frame interlacing technique in conjunction with liquid crystal FE-1 goggles (Cambridge Research Systems, Cambridge, UK) synchronized with the monitor’s frame rate, and were updated in real time with head and hand movements (acquired on-line with sub-millimeter resolution by using an Optotrak Certus with one position sensor) so to keep their geometrical projection always consistent with the participant’s viewpoint.

Figure 3 Temporal sequence of phases in our Experiments (A) and details of the facial emotion detection task (B).

(A) Two blocks of Experiment 1 separated by the calibration phase. In the first induction phase the valence of the motor action is negative, and the test phase requires the detection of a face displaying an incongruent positive emotion. In the second induction phase the valence of the motor action is instead positive, and the test phase requires the detection of a face displaying a congruent positive emotion. The inset in (B) shows the detailed temporal sequence of events in one trial of the test phase (facial emotion detection task).

Each reaching session was preceded by the same calibration procedure used in Fantoni & Gerbino (2014), and detailed in Nicolini et al. (2014). In particular, the positions of the index tip and of the eyes (viewpoint) were calculated during were calculated during the system calibration phase using three infrared-emitting diodes firmly attached on the distal phalanx and on the back of the head. This was needed to ensure a correct geometrical projection of the 3D visual displays (virtual rod and index tip) according to the shifts of different body segments. Specifically, head movements updated the participant’s viewpoint to present the correct geometrical projection of the stimulus in real time. A custom made Visual C++ program supported stimulus presentation and the acquisition of kinematic data associated to the reaching phase, as well as the recording of yes/no responses relative to the facial emotion detection task (left/right keys of the computer keyboard) and RTs.

The simulated egocentric depth of the rod along the line of sight was manipulated according to empirical data by Fantoni & Gerbino (2014), but see also Mark et al. (1997). Fantoni & Gerbino (2014) asked observers to rate the discomfort of 50 reaches whose depth was randomly varied across trials in the entire [0.65–1.00] range of arm length using a 0–50 discomfort scale adapted from the pain scale by Ellermeier, Westphal & Heidenfelder (1991). Following their results the simulated egocentric depth of the rod axis along the line of sight was randomly chosen within the [0.65–0.75] range for the comfortable-reaching block and within the [0.90–1.00] range for the uncomfortable-reaching block, relative to the arm length of each participant.

Furthermore, a physical rod (equal in shape to the virtual one) placed behind the mirror that fully occluded it was attached to a linear positioning stage (Velmex Inc., Bloomfield, NY, USA). The position of the physical rod was matched to the egocentric depth of the simulated rod on a trial-by-trial basis, with submillimiter precision (straight-line accuracy = 0.076 mm) with the Velmex motorized Bslides assembly (Bloomfield, NY, USA), so that real and virtual stimuli were perfectly aligned. This provided participants with a fully consistent haptic feedback as the red sphere marking the index tip reached the illusory surface defined by the constellation of random dots shaping the virtual rod exactly when the participant’s finger entered in contact with the real rod. Furthermore to ensure consistent vergence and accommodative information, the position of the monitor was also attached to a Velmex linear positioning stage that was adjusted on a trial-by-trial basis to equal the distance from the participant’s eyes to the virtual/real object that should be reached during the reaching block. Synchronizing stimulus presentation with the motorized linear positioning stage system allowed us to randomly manipulate the distance of the reaches in a rather continuous way (step resolution = 0.076 mm) over the depth ranges used in our study; i.e., in both comfortable ([0.65–0.75] the arm length) and uncomfortable ([0.75–1.00] the arm length) reaching blocks.

The facial stimulus set included the same 8 characters (four Caucasian males and four Caucasian females) that Fantoni & Gerbino (2014) selected from the Radboud University Nijmegen set (Langner et al., 2010): namely, female models 1, 4, 14, 19; and male models 20, 30, 46, 71. For each of the 8 selected characters of the Radboud database (included in Fantoni & Gerbino (2014), Fig. 1) we utilized color photographs displaying faces expressing two basic emotions, happiness and anger, and the corresponding neutral faces (all stimuli produced a high agreement with intended expressions in the validation study). A neutral-to-angry and a neutral-to-happy continua were first generated for each of the 8 characters, morphing the fully angry/happy face and the neutral face in variable proportions, in 5 per cent steps, using MATLAB software adapted from open source programs. For every two pairs of facial images we selected about 75 key points. The software generated a synthetic image containing a specified mixture of the original expressions, using a sophisticated morphing algorithm that implements the principles described by Benson & Perrett (1999). As in Marneweck, Loftus & Hammond (2013) and Fantoni & Gerbino (2014), we identified corresponding points in the two faces, with more points around areas of greater change with increasing emotional intensity (pupils, eyelids, eyebrows, and lips). Then, as depicted in Fig. 1B, for every character we selected three target stimuli for the neutral-to-angry continuum and three target stimuli for the neutral-to-happy continuum, corresponding to the following morph intensities: 10% emotion (90% neutral), 20% emotion (80% neutral), 30% emotion (70% neutral). Neutral stimuli were the original “no emotion” faces of the 8 selected characters. All images were aligned for facial landmarks and masked by an oval vignette hiding hair and ears presented on a black surround, the vignette being centered on the screen and occupying a visual size of 7.5° × 10.7° at the viewing distance of 50 cm.

The target emotion was positive (happiness) in Experiments 1 & 2 vs. negative (anger) in Experiments 3 & 4. Each experiment included two random sequences (one for each MAMIP condition, comfortable vs. uncomfortable) of 32 facial images resulting from the product of 8 characters × 4 morph intensities (including the neutral face corresponding to 0% emotion in the morph).

The complete 2 × 2 × 2 × 4 mixed factorial design shown in Fig. 1A included the two between-subjects factors called Action Ordering (comfortable ⇒ uncomfortable vs. uncomfortable ⇒ comfortable, Fig. 1A, top row) and Target Emotion (positive vs. negative, Fig. 1A, matrix rows) and the two within-subjects factors called action/emotion valence Congruency (congruent vs. incongruent, Fig. 1A, matrix columns, depending on encoding in each cell) and Morph Intensity (0, 10, 20, 30% emotion in the morph, Fig. 1B).

Preliminary experiment

The values of morph intensities used for the target stimuli in our two emotion continua were established empirically on the basis of the results of the preliminary baseline experiment. The instructor (author WG) explained that in each trial two faces (one neutral and one displaying an emotion) would be presented (1500 ms exposure) on the left/right of a continuously visible central fixation cross. He then explained that the positions of the two faces were randomized and that the task consisted in writing the letter S (sinistra, left) or D (destra, right) in the appropriate cell of the response sheet, to indicate the position of the face expressing the emotion. The response sheet required participants to answer preliminary questions about their gender and age, and to fill 32 cells (16 for each of the two experimental blocks).

The instructor then presented a sample of 8 trials, to familiarize participants with all characters: four practice trials included a morphed happy face and a morphed angry face. The experimental session included two blocks of 16 trials each. The first block included a random sequence of trials in which a neutral face was shown together with a 15% happy face (8 characters by two positions); the second block included a different random sequence of trials in which a neutral face was shown together with a 15% angry face (always balancing characters and positions). To prevent mistakes in filling in the response sheet, the instructor named the trial number aloud, before the presentation of the 300 ms fixation dot preceding stimulus presentation.

Stimuli were presented using PowerPoint through a high resolution MARCA video projector connected to the graphic output of MAC-PRO (3D graphic accelerator). Participants were comfortably seated in a dimly lit classroom while facing the projection screen at the average distance of 12.25 m away. The average visual angle subtended by classroom displays was similar to the visual angle in Experiments 1–4, given that they were 35 times larger than the stimuli displayed on the lab CRT and the participant’s distance from the projection screen was about 35 times the one in the lab.

We extracted individual d′ values from raw proportions of “left” responses out of 8 trials, for both hits and false alarms, using a glm with variable intercept β1 and slope β2 for every participant and emotion condition (Knoblauch & Maloney, 2012). This corresponded to reparametrize each individual Gaussian function fit in term of β2/√2, or the d′ for a 2AFC paradigm (i.e., the difference between hit and false alarm rates on the probit scale).

From the analysis of valid d′ values (those between ±2.5 individual standard deviation from the mean which led to the removal of 5 d′ values from the negative and 4 d′ values from the positive target emotion condition, each out of the 91 values collected over the two entire emotion conditions) two main results are: (1) the 15% morph level produced a sizable perceptual effect on emotion detection eliciting a non null sensitivity for both positive (1.26 ± 0.159, two-tailed t vs. 0 = 7.90, df = 86, p < 0.001, d = 1.70) and negative (0.15 ± 0.049, two-tailed t vs. 0 = 2.96, df = 85, p = 0.0039, d = 0.64) target emotions; (2) emotion detection as elicited by our stimulus set was anisotropic as revealed by the lower average sensitivity in the anger rather than in the happiness detection task (Welch two sample t = 6.64, df = 171, p < 0.001, d = 1.01).

These preliminary results were in agreement with previous results in the emotion perception literature showing that realistic faces, as those used in the current experiments, often give rise to a happiness (rather than the more often found anger) detection advantage relative to both angry (Becker et al., 2011; Becker et al., 2012; Juth et al., 2005) and sad (Srivastava & Srinivasan, 2010) faces. Furthermore, the results demonstrated that morphing from our face set in a range above and slightly below the 15% should elicit sizable sensitivity differences with both positive and negative emotions, thus setting the optimal conditions for measuring the effect of bodily action on the perception of emotion. Finally, the average d′ values obtained with such a large sample of observers is representative of emotion detection performance in a baseline inaction condition and can thus be used as a reference value to evaluate whether, according to G3 (see also Fig. 2), the effect of MAMIP is symmetric or not (results in Fig. 7).

Procedure

As shown in Fig. 3, our procedure included two sessions, each composed by an induction phase (MAMIP reaching phase) followed by a test phase (emotion detection task).

MAMIP reaching phase

To maximize data comparability the present study followed the original procedure implemented by Fantoni & Gerbino (2014); (see their procedure section for details). Participants were asked to perform 50 successive unrestrained reaches towards a virtual random-dot rod positioned along the line of sight at a variable depth randomly selected with submillimiter precision (0.076 mm) within the [0.65–0.75] range of individual arm length for the Comfortable session and within the [0.90–1.00] range of individual arm length for the Uncomfortable session. According to Mark et al. (1997), leaving unrestraining the body during reaches allows the number of degrees of freedom involved in the motor act to vary consistently with the depth of the reach that, in turn, should be a prerequisite for the establishment of a concurrent variation of comfort/discomfort. Participants were instructed to perform the movement in a rather natural way (neither too slowly nor to fast), to rest with their index in contact with the real rod until the disappearance of the stimulus, and then to move back their hand in the starting position.

The finger movement started in full darkness from a fixed out-of-view position shifted relative to the body midline by about 25 cm from the sagittal plane and 15 cm from the coronal plane. This position was registered at the end of the system calibration phase after which the linear positioning stages for the monitor and for the real rod moved for about 1000 ms. At the end of such positioning phase an acoustic feedback (200 Hz, lasting 200 ms) signalled that the observer could start her movement. The virtual rod and virtual red sphere marking the fingertip were visible for about 3.5 s from the moment the finger entered in the participant’s visual field after a shift of about 3 cm from its starting position. The end of the reaching movement was accompanied by a haptic feedback and was followed (after a variable lapse of time depending on individual reaching velocity) by the same acoustic signal after which the simulated rod disappeared for about 1000 ms. As soon as the simulated rod disappeared the observer was asked to place her hand again in the starting position. At the end of the blank period, the motorized positioning stages were activated so to adjust the position of the physical rod behind the mirror to the egocentric depth of the successive simulated depth rod. At the end of the motor movements (after 3000 ms) a second acoustic signal (200 ms) was provided to the observer signalling that she could start the hand movement. Successful reaches were ensured by providing the observer with a fully consistent haptic feedback at the end of each reaching act. This was possible thanks to the motorized positioning system used to perfectly align on a trial by trial basis the position of the physical rod to the one of the simulated rod. Furthermore, reaching kinematics and head position were on-line monitored by the experimenter from a remote station invisible to the observer.

Each reaching session lasted a total of 8 min on average, subdivided as follows: 4.3 min of forward reaching actions +0.83 min of backward reaching actions +2.83 min dead times (including the time of motion of linear positioning stages and of acoustic signals). The forward reaching acts can be described as composed by three successive phases: (1) a movement planning phase (from the acoustic signal to the appearance of the virtual random dot, lasting 1.67 ± 0.18 s); (2) an execution phase (from the appearance of the virtual random dot to the time of index contact with the real rod, lasting 2.09 ± 0.07 s for long uncomfortable and 1.59 ± 0.06 s for short comfortable reaches); (3) an exploration phase (from finger contact until the disappearance of the object, lasting 1.41 ± 0.07 s for long uncomfortable and 1.91 ± 0.06 s for short comfortable reaches).

The procedure included: a session in which the participant’s arm length at rest (i.e., the effective maximum reach) was carefully measured following a procedure similar to the one used by Mark et al. (1997; Appendix 1A), instructions, a training with 15 reaches randomly extracted across the entire depth range used in the experiment (0.65–1.00 of arm length), and the experimental session. We decided to eliminate participants that during the training session were unable to perform a sequence of at least 5 successful reaches (getting in contact with the real rod within the predefined temporal interval of 3.5 s) in the last 8 trials. No participants were excluded on the basis of this action criterion.

Facial emotion detection (yes/no task) phase

Participants completed a randomly ordered block of 32 facial emotion detection trials just after each reaching session. Each block resulted from the combination of 8 characters (4 actors and 4 actresses) × 4 morph intensities (0, 10, 20, 30% emotion in the morph), with an overall 3:1 ratio of [S + N] to [N] trials. In other terms, every stimulus was presented to participants only once, to minimize possible attempts to reproduce responses already given to stimuli remembered as identical to the target and to keep the test phase reasonably short, compared to duration of the induction phase. The target emotion was happiness in Experiments 1 & 2 and anger in Experiments 3 & 4.

The same yes/no task was applied in all four experiments. At the beginning of each emotion detection trial a 30-pixel-wide green fixation circle was displayed at the center of the screen for about 500 ms (Fig. 3B). This was substituted by a brief refreshing blank screen of about 150 ms. The face stimulus was then displayed until the participant pressed one of two response keys with his/her left hand: left key for yes (“Emotion present”) vs. right key for no (“Emotion absent”). The default minimum duration of the face stimulus was 400 ms. After key press, a low tone lasting 400 ms signalled the response recording and a blank screen lasting about 850 ms followed. The end of such a blank screen period was signalled by a mid tone acoustic feedback lasting 200 ms. The next trial was thus presented. Notice that the left hand was used for responses to the emotion detection task while the right hand, wearing markers, was resting after the reaching session.

On average each facial emotion detection session lasted a total of 1.6 min subdivided as follows: 0.6 min of response/observation time (32 trials × average OT = 1154 ± 14 ms) + 1.0 min of rest times (including the timing of fixation, acoustic signals and blank screen period).

The procedure included: (a) instructions; (b) a session of familiarization with the face set (including a serial presentation of the 32 facial stimuli to be presented in the facial emotion detection task, ordered by character and morph intensity); (c) a training block of 32 emotion detection trials (in which the face of each of the 8 characters was presented four times, one in the neutral pose and three times as a 50% morph of the emotion appropriate for each experiment); (d) the experimental session. The training block was designed having in mind two goals: (a) familiarization with neutral faces, [N] trials, and with the target emotion, when its intensity was well above the levels utilized in [S + N] trials; (b) elimination of participants with an inadequate level of sensitivity. Only participants with more than 90% correct responses during training entered the experimental session (four participants excluded). Written instructions required participants to use the green circle to support steady fixation during stimulus presentation, to keep in mind that just one fourth of the stimuli displayed a neutral facial expression, and to be fast and accurate, considering that stimulus presentation was terminated by the response. We chose this option, rather than a fixed exposure time, to account for individual variability in the processing of different emotions and to allow observers to modulate the amount of time in which stimulus-driven emotional information was available. The response-terminated presentation method sets the conditions for: (1) a trade-off between individual d′ and response/Observation Time (OT); (2) an inverse modulation of OT as a function of the intensity of the signal in [S + N] trials (Fantoni, Gerbino & Kellman, 2008; Gratton et al., 1988; Wickens, 1984).

We ran a preliminary lme analysis on the relationship between individual d′ values (see the Statistical analysis section for details on their computation) and response speed, computed as the inverse of OT values for correct responses (i.e., 1000/OT in the interval between ±2.5 SD from the individual mean, which led to the removal of 27 out of 2496 total trials). Response speed values were averaged within all cells of the overall experimental design. The lme analysis revealed: (1) a weak speed-sensitivity trade-off, given that the main effect of response speed on d′ was marginal (F1,224.6 = 3.607, p = 0.060), though it increased as a function of per cent emotion in the morph (F1,201.1 = 6.330, p = 0.012); (2) the response speed increased at a rate of 0.13192 ± 0.01349 s every 10% increment in the morph (F1,195 = 95.637, p < 0.001), independently from other experimental factors like action/emotion Congruency (F1,195 = 0.038, p = 0.84), Action Ordering (F1,39 = 0.98, p = 0.32) and Target Emotion (F1,39 = 1.45, p = 0.235). The lack of interaction between OT (i.e., response time) and experimental factors supports our decision to focus the following analysis on indices of signal detection performance.

Results and Discussion

Statistical analyses

Following Knoblauch & Maloney (2012), all indices of signal detection performance (both perceptual and decision based) were calculated by applying a generalized linear model (glm) with a probit link function to the whole set of binary responses. Individual triplets of d′ values associated to the three combinations of hits (yes responses to 10, 20, 30% morph, respectively) and false alarms (yes responses to 0% morph) were extracted using a glm with variable intercept β1 and slope β2 for every participant, reaching session, emotion, and action ordering. We encoded signal presence/absence as a discrete variable (1 for trials with morph intensity >0; 0 for trials with the original neutral face) and reparametrized each individual Gaussian function fit in term of its slope (corresponding to the difference between hit and false alarm rates on the probit scale), or d′.

Generalizing upon this statistical technique and following Marneweck, Loftus & Hammond (2013) we further extracted two global indices of detection performance by fitting the same glm to the entire set of yes responses as a function of signal presence/absence. The two indices were global d′ for the perceptual component of the model and response criterion c [as given by −(2β1+β2)2] for the decision component. The c index provides a measure of response bias independent of sensitivity to facial expressions of emotion, as needed to draw conclusions relevant to our second goal (G2). Individual c values indicate how far the criterion used by the observer to deliver a yes response departs from the optimal decision rule (i.e., equal false alarm and miss rates), with negative c values indicating an unbalance in favor of yes over no responses. On the average, a negative c value was expected as a consequence of the unbalanced [S + N]/[N] ratio (with only 1/4 of trials displaying a neutral face).

To provide an additional measure of a possible mood-congruent effect on sensitivity to facial emotions, revealed by our detection task, we also analyzed AT values as derived from intercept and slope values estimated by a linear mixed-effect (lme) model with participants as a random effect and morph intensity as the continuous covariate of individual d′ triplets applied to each of the four experimental groups of participants separately.

The negative sign of the slope between individual d′ triplets and per cent target emotion was used as exclusion criterion, given that AT values computed from an indirect relationship between d′ and morph intensity were statistically meaningless: this led to the removal of one participant from the analysis of data from Experiment 1 (out of the total of 40).

Distributions of individual values of performance indices d′, global d′, AT, c were analyzed using a step-wise procedure that contrasted linear mixed-effect (lme) models of increasing complexity (Bates et al., 2014), depending on the number of fixed effects, modelled by the factors of our experimental design (action/emotion valence Congruency, Target Emotion, and Action Ordering). Models were fitted using Restricted Maximum Likelihood. Participants were treated as random effects so to control for the individual variability of emotion detection performance. We followed Bates (2010) and used this statistical procedure to obtain two-tailed p-values by means of likelihood ratio test based on χ2 statistics when contrasting lme with different complexities (for a discussion of advantages of a lme procedure over the more traditional mixed models analysis of variance see Kliegl et al., 2010). We used type 3-like two tailed p-values for significance estimates of lme’s fixed effects and parameters adjusting for the F-tests the denominator degrees-of-freedom with the Satterthwaite approximation based on SAS proc mixed theory. Among the indices that have been proposed as reliable measures of the predictive power and of the goodness of fit for lme models we selected the concordance correlation coefficient rc, which provides a measure of the degree of agreement between observed and predicted values in the [−1, 1] range (Vonesh, Chinchilli & Pu, 1996; Rigutti, Fantoni & Gerbino, 2015). Post-hoc tests were performed using paired two sample t-tests with equal variance. As measures of significant effect size, depending on the statistical analysis, we provided Cohen’s d, the coefficient of determination r2, and/or rc.

Evidence from the distributions of yes responses

Figure 4 illustrates the average percentages of yes (“Emotion present”) responses (and SEMs) together with the best fitting cumulative Gaussian (averaged across participants as modelled through glm) as a function of per cent emotion for the two levels of action/emotion Congruency: congruent (red) vs. incongruent (cyan). Panels A, C show detection data from Experiments 1 and 3, following uncomfortable-comfortable MAMIP sessions; panels B, D show detection data from Experiments 2 and 4, following comfortable-uncomfortable MAMIP sessions. Emotions to be detected were happiness for Figs. 4A and 4B and anger for Figs. 4C and 4D.

Figure 4 Modeling the distributions of “Emotion present” percentages with glm.

The four panels show the average percentages of yes (“Emotion present”) responses ± SEM as a function of per cent emotion, when the target emotion was congruent/incongruent (red/cyan symbols, respectively) with action/mood valence. Red/cyan curves are the best average cumulative Gaussian fits of response percentages, with shaded bands indicating ± standard error of regression. Action ordering is illustrated by the legend on top. Top panels A, B refer to happiness detection ((A) Experiment 1, uncomfortable ⇒ comfortable; (B) Experiment 2, comfortable ⇒ uncomfortable); bottom panels C, D refer to anger detection ((C) Experiment 3, uncomfortable ⇒ comfortable; (D) Experiment 4, comfortable ⇒ uncomfortable).

A preliminary statistical analysis revealed that the glm procedure used to extract our indices of detection performance provided a very good fit to our dataset and was robust enough to describe the metric of responses obtained in our four experiments. In all tested conditions (Experiments 1–4) the best linear fit describing the relationship between individual predicted and individual observed yes percentages was a line with unitary slope and null intercept accounting for a large percentage of variance, as shown in Table 1.

Table 1 Summary table of yes distributions in Experiments 1–4.

Per cent explained variance and statistical indices of goodness of fit of glm-based vs. observed yes distributions in Experiments 1–4.

			Best glm fitting parameters	
	Per cent explained variance	Goodness of fit	β1	β2	
Experiment 1 (uncomfortable ⇒ comfortable)	86	F(1, 78) = 504.0, p < 0.001	0.002 ± 0.035, t = −0.059, p = 0.95	1.00 ± 0.045, t vs. 1 = 0.02, p = 0.98	
Experiment 2 (comfortable ⇒ uncomfortable)	88	F(1.78) = 595.7, p < 0.001	0.005 ± 0.031, t = 0.17, p = 0.86	0.99 ± 0.040, t vs. 1 = 0.18, p = 0.85	
Experiment 3 (uncomfortable ⇒ comfortable)	86	F(1, 78) = 551.9, p < 0.001	0.033 ± 0.030, t = −1.07, p = 0.28	1.05 ± 0.044, t vs. 1 = 0.77, p = 0.44	
Experiment 4 (comfortable ⇒ uncomfortable)	73	F(1, 70) = 185.2, p < 0.001	0.001 ± 0.05, t = 0.17, p = 0.87	0.98 ± 0.070, t vs. 1 = 0.15, p = 0.87	

The graphs in Fig. 4 clearly show the effect of action/emotion congruency on sensitivity. The pattern of responses is fully consistent with the facilitation-by-congruency hypothesis and at odds with both a facilitation-by-arousal hypothesis and explanations based on action valence per se. In all panels the increase of yes responses as a function of per cent emotion is well described by two positive halves of a sigmoid, with the red curve fitting the data from the block in which the valence of the target emotion was congruent with the valence of the reaches (Figs. 4A and 4B: happiness detection after comfortable reaches; Figs. 4C and 4D: anger detection after uncomfortable reaches) vs. the cyan curve fitting the data from the block in which the valence of the target emotion was incongruent with the valence of the reaches (panels A, B: happiness detection after uncomfortable reaches; panels C, D: anger detection after comfortable reaches).

Consistently with the facilitation-by-congruency hypothesis, yes percentages monotonically increased with morph intensity, with the rate of increase being larger (indicating higher sensitivity) for the red mood-congruent curve than the cyan mood-incongruent one, in all tested conditions: the yes percentage was indeed smaller in the mood-congruent (red points) than mood-incongruent (cyan points) conditions for small (determining an overall lower false alarm rate) but not large values of per cent emotion in the morph (determining an overall higher hit rate), in both anger and happiness detection tasks. This was confirmed by the results of the lme analysis revealing that the pattern of average “happy” percentages (t = 31.7, df = 158; p = 0.00; r2 = 0.86, 95% CI [0.82, 0.90], rc = 0.93, 95% CI [0.90, 0.95]) determined a significant main effect of Morph Intensity (F3,140 = 278.4, p < 0.001), and a significant Morph Intensity × Congruency interaction (F3,140 = 4.078, p = 0.008). The pattern of average “angry” percentages was similarly distributed (t = 31.7, df = 158; p = 0.00; r2 = 0.80, 95% CI [0.74, 0.85], rc = 0.88, 95% CI [0.85, 0.91]), though not determining a fully significant Morph Intensity × Congruency interaction (F3,133 = 2.10, p = 0.10), probably because of the overall noisier pattern of responses: the consistency of “angry” percentages (average individual variability of yes percentages quantified by average standard error of the mean values) being worse in the anger (0.0563) than in the happiness (0.035) detection task (t = 2.867, df = 30, p = 0.007, d = 1.04). The interaction patterns rising from both distributions of yes percentages were due to the negative congruency gain (Per cent yes congruent–Per cent yes incongruent) at 0% emotion (Experiments 1 and 2, Figs. 4A and 4B: −0.125 ± 0.034, t = −3.649, df = 140, p < 0.001; Experiments 3 and 4, Figs. 4C and 4D: −0.088 ± 0.040, t = −2.221, df = 133, p = 0.028), combined with a positive or null gain for positive at 10–30% emotion.

Notably, the higher sensitivity for detecting happiness after comfortable rather than uncomfortable reaches (Experiments 1 and 2, Figs. 4A and 4B) is consistent with facilitation-by-congruency but not facilitation-by-arousal, given that an arousal-based improvement in emotional face processing should occur after the uncomfortable MAMIP session (requiring a higher level of motor activation/arousal than the comfortable). Furthermore, the pattern of yes responses depicted in Fig. 4 rules out any explanation based on action valence per se. Reaches of opposite valence led to similar improvements of emotion detection performance, consistently with action/emotion valence congruency: in Experiments 1 and 2 (Figs. 4A and 4B) happiness detection was improved after a comfortable (not uncomfortable) MAMIP session; while in Experiments 3 and 4 (Figs. 4C and 4D) anger detection was improved after an uncomfortable (not comfortable) MAMIP session. Confirming previous results (Fantoni & Gerbino, 2014), detection of facial emotions was improved by the congruency between the valence of the inducing actions and the valence of the target emotion.

Evidence from the distributions of sensitivities, thresholds and response bias

Conclusions from the previous analysis on yes percentages were corroborated by lme statistics on indices of detection performance. These quantitative analyses also allowed us answering two major questions about the facilitation-by-congruency effect induced by MAMIP: (1) is it an additive or multiplicative effect?; (2) is it a perceptual or post-perceptual effect?

Action/emotion valence congruency improves happiness detection but not response bias: against arousal

Figure 5 shows the average d′ triplets (Figs. 5A, 5E), AT (Figs. 5B, 5C, 5F, 5G) and c (Figs. 5D, 5H) for the two reaching sessions in Experiments 1 (Figs. 5A, 5B, 5C, 5D, uncomfortable ⇒ comfortable) and 2 (panels E, F, G, H, comfortable ⇒ uncomfortable). Independent of Action Ordering (panels A vs. E) the distributions of average d′ values for happiness detection as a function of per cent target emotion were in strong agreement with an additive (not multiplicative) facilitation-by-congruency effect, as confirmed by the lme analyses of Experiments 1 and 2, with participants as random effects, and emotion Congruency (comfortable action ⇒ happy expression vs. uncomfortable action ⇒ happy expression) and Morph Intensity (10, 20, 30% emotion) as fixed effects. A separate analysis of data from Experiments 1 and 2 follows.

Figure 5 Action/emotion valence congruency affects happiness detection sensitivity, but not response bias.

Average d′ triplets (A, E) and average ATs (B, F), for the congruent (comfortable, in red) and incongruent (uncomfortable, in cyan) reaching sessions (B, F), together with individual ATs (C, G) and response biases (D, H) changes due to congruency, in Experiments 1 (uncomfortable ⇒ comfortable) and 2 (comfortable ⇒ uncomfortable) as coded by the icons on top. Error bars represent ± SEM. Red and cyan lines in panels A, E are the lme model regression line, with the shaded region corresponding to ± standard error of the regression. The spatial arrangement of these lines in the d′ by per cent emotion graph is informative about the influence of action/emotion valence congruency on happiness detection, with a facilitation-by-congruency effect signaled by the red line being above the cyan line and with their parallelism being diagnostic of an additive (not multiplicative) effect. (B, F) A red point closer to zero than the cyan indicates a lowering of the absolute threshold for happiness induced by congruency. (C, G) Individual threshold changes between incongruent and congruent reaching sessions in Experiments 1 and 2, respectively. A negative change represents an increased likelihood that a smaller value of per cent happiness in the morph would elicit a yes response after the incongruent than congruent session. The vertical green line represents the grand average threshold change due to congruency ± SEM. (D, H) Individual response bias changes between incongruent and congruent reaching sessions in Experiments 1 and 2, respectively. A negative change represents an increased likelihood towards a positive bias after the congruent than incongruent session. The vertical green line represents the grand average response bias change due to congruency ± SEM.

In Experiment 1 (Fig. 5A), happiness detection increased linearly with an lme estimated rate of about 2.6 ± 0.26 d′ units every 10 per cent increment in the morph (F1,48 = 138.56, p < 0.001), and with a constant d′ increment of about 0.84 ± 0.36 units in congruent (comfortable-happy) over incongruent (uncomfortable-happy) conditions (F1,48 = 5.49, p = 0.023). Only 50 reaching acts distributed over 10 min, with a slightly different depth extent (average depth difference between comfortable and uncomfortable reaches = 17.56 cm ± 0.18), produced systematic changes in the detection of subtle variations of happiness. This is consistent with the idea that reaches with positive (i.e., comfortable) but not negative (i.e., uncomfortable) valence pre-activate detectors selectively tuned to emotional facial features with the same valence (i.e., happiness), which in turn facilitate the performance in Congruent relative to Incongruent conditions.

Remarkably, no improvement of fit was found (χ12=0.18, p = 0.67) with a separate slope lme including the Morph Intensity × Congruency interaction (F3,56 = 43.44, p < 0.001, r2 = 0.74, rc = 0.85, 95% CI [0.76, 0.90]), relative to an equal slope lme not including the interaction (F2,57 = 66.05, p < 0.001, r2 = 0.74, rc = 0.85, 95% CI [0.76, 0.90]). This is a proof that the facilitation-by-congruency effect produced by MAMIP on happiness detection followed an additive rather than multiplicative trend in the d′ by morph intensity space.

This is corroborated by the post-hoc analyses (one tailed paired t test) showing that any 10% increment in morph intensity (i.e., from 10 to 20 or from 20 to 30) produced a significant d′ gain (see Table 2). These d′ gains were similar in magnitude in incongruent-uncomfortable (t = 0.78, df = 9, p = 0.45, d = 0.52) and congruent-comfortable conditions (t = 0.78, df = 9, p = 0.45, d = 0.52).

Table 2 Summary of post hoc analyses for Experiments 1 and 2.

		Per cent happiness range	Mean d′ gain and SEM	Paired t test	
Experiment 1	Incongruent	[10%–20%]	2.41 ± 0.65	t = 3.80, df = 9, p = 0.004, d = 2.53	
[20%–30%]	2.57 ± 0.73	t = 3.94, df = 9, p = 0.003, d = 2.62	
Congruent	[10%–20%]	2.77 ± 0.75	t = 4.11, df = 9, p = 0.003, d = 2.76	
[20%–30%]	2.58 ± 0.81	t = 3.62, df = 9, p = 0.005, d = 2.23	
Experiment 2	Incongruent	[10%–20%]	3.37 ± 0.82	t = 4.51, df = 9, p = 0.001, d = 3.00	
[20%–30%]	2.15 ± 0.81	t = 2.86, df = 9, p = 0.019, d = 1.90	
Congruent	[10%–20%]	3.47 ± 0.72	t = 5.32, df = 9, p < 0.001, d = 3.55	
[20%–30%]	2.06 ± 0.77	t = 3.09, df = 9, p = 0.013, d = 2.06	

The additive facilitation produced by MAMIP on happiness detection performance is further corroborated by the significant facilitation-by-congruency effect on: (1) global happiness sensitivity (F1,9 = 9.46, p = 0.01, d = 1.94), which was larger after the comfortable (global d′ = 1.76) than uncomfortable (global d′ = 1.20) reaching session, 95% CI [0.15, 0.97]; (2) the absolute threshold for happiness (F1,9 = 5.79, p = 0.03, d = −1.52), given that, as shown in Fig. 5C, the AT value was smaller after participants completed a comfortable (AT = 4.6 ± 1.2%) than uncomfortable (8.1 ± 1.4%) MAMIP session, 95% CI [0.21, 6.77].

As shown in Fig. 5D, the response bias was clearly influenced by the unbalanced [S + N]/[N] trial ratio. As expected, the 3:1 ratio produced an overall negative bias; i.e., a dominance of “Emotion present” over “Emotion absent” responses in both Congruent (−0.31 ± 0.13, t = −2.714, df = 15.04, p = 0.016, d = −1.40) and Incongruent conditions (−0.52 ± 0.10, t = −4.603, df = 15.04, p < 0.001, d = −2.37). However, unlike sensitivity, the bias was independent of action valence (comfortable vs. uncomfortable). To control for the perceptual vs. post-perceptual locus of the MAMIP effect, according to G2, the same lme analysis conducted on perceptual indices of happiness detection (global d′ and AT) was conducted on response criterion c. The MAMIP effect was not replicated. The main effect of Congruency on c was non significant (F1,9 = 3.0, p = 0.12).

As depicted in Figs. 5E, 5F, 5G and 5H, the facilitation-by-congruency effect was strikingly similar in Experiment 2 (uncomfortable ⇒ comfortable). Again, the pattern of happiness detection performance shown in Fig. 5E was optimally accounted for by an equal slope lme (F2,57 = 71.85, p < 0.001, r2 = 0.79, rc = 0.88, 95% CI [0.81, 0.92]), consistent with an additive facilitation-by-congruency effect induced by MAMIP, not by a separate slope lme (F3,56 = 47.06, p < 0.001, r2 = 0.79, rc = 0.88, 95% CI [0.81, 0.92]), consistent with a multiplicative facilitation by congruency effect induced by MAMIP (χ12=0.0, p = 0.99). The lme analysis revealed a similar, though not significant (F1,48 = 1.46, p = 0.23), constant increment of sensitivity due to action/emotion valence congruency (0.23 ± 0.30), and a similar linear modulation of sensitivity by the per cent happiness in the morph (β2 = 2.76 ± 0.20, F1,48 = 179.96, p < 0.001). This was confirmed by post-hoc paired t-tests. As in Experiment 1, any 10 per cent increment in morph intensity (i.e., from 10 to 20 or from 20 to 30) produced a significant d′ gain (see Table 2), with the d′ gains being similar in magnitude in incongruent-uncomfortable (t = 0.78, df = 9, p = 0.45, d = 0.52) and congruent-comfortable conditions (t = 0.78, df = 9, p = 0.45, d = 0.52). Finally the average difference between d′ values for congruent and incongruent conditions was almost constant at increasing per cent happiness in the morph, with the performance gain due to congruency for 10, 20, and 30 per cent happiness in the morph being equal to: 0.20 ± 0.14 (t = 1.81, df = 9, p = 0.09, d = 1.20), 0.29 ± 0.087 (t = 3.83, df = 9, p = 0.004, d = 2.55), and 0.20 ± 0.088 (t = 2.68, df = 9, p = 0.025, d = 2.68), respectively.

The different facilitation-by-congruency effect sizes revealed by the lme analyses in Experiments 1 and 2 were likely due to the unbalanced temporal ordering of reaching sessions. Participants in Experiment 2 were indeed less experienced with both the detection task and the face set after the comfortable (congruent) than uncomfortable (incongruent) MAMIP session, given that the comfortable condition occurred first.

Surprisingly, despite the fact that the effects of action/emotion valence congruency and learning were in opposite directions in Experiment 2, thus reducing the performance difference induced by the two reaching sessions, we found the same significant facilitation-by-congruency effect observed in Experiment 1 on both global d′ (F1,9 = 8.09, p = 0.02, d = 1.79) and AT (F1,9 = 8.45, p = 0.02, d = 1.84), even in the absence of significant shifts in response bias c (F1,9 = 0.96, p = 0.37). As shown in Figs. 5F and 5G, action/emotion valence congruency increased global d′ by about 0.24 d′ units, 95% CI [0.05, 0.43], and decreased AT by about 0.83 per cent emotion, 95% CI [0.05, 0.43]. Again, similar, though negative, c values were obtained in congruent (−0.32 ± 0.10, t = −2.85, df = 11.52, p = 0.015, d = −1.68) and incongruent (−0.40 ± 0.12, t = −3.53, df = 15.04, p = 0.004, d = −2.10) conditions (Fig. 5H), with an average response bias change due to congruency of about (−0.077, 95% CI [−0.26, 0.10]).

Is there an effect of Action Ordering on happiness detection?

To test the possible role of Action Ordering we compared the patterns of d′, AT and c in Experiment 2 directly to those of Experiment 1, including Experiment as an additional fixed effect in lme analyses.

In a first lme analysis we thus asked how the relationship between individual d′ values and morph intensities were affected by Congruency and/or Action Ordering. In this model (t = 19.66, df = 118, p = 0.00, r2 = 0.77 95% CI [0.68, 0.83], rc = 0.86, 95% CI [0.81, 0.90]) d′ resulted to be positively affected by Morph Intensity (β2 = 0.27 ± 0.015, F1,98 = 315.19, p < 0.001) and Congruency (estimated d′ gain = 0.53 ± 0.24, F1, 98 = 4.73, p = 0.03): other main effects or interactions were not statistically significant (χ52=3.08, p = 0.68), with the intercept of the equal slope lme model, for the mood congruent condition, being negative (β1 = −1.49 ± 0.37, df = 113.56, t = −4.02, p < 0.001, d = 0.754). A second lme model on global d′ (t = 10.60, df = 38, p = 0.00, r2 = 0.75, 95% CI [0.57, 0.83], rc = 0.79, 95% CI [0.68, 0.86]) revealed a significant main effect of facilitation by congruency of about 0.4 ± 0.1 d′ units (F1,18 = 15.94, p < 0.001); while neither the effect of Experiment (F1,18 = 0.0020, p = 0.96) nor the Congruency × Experiment interaction (F1,18 = 2.49, p = 0.13) were significant. Similar results were obtained on ATs, given that facilitation-by-congruency, again, resulted to be the only significant factor affecting the performance (estimated AT after comfortable-congruent = 5.24 ± 0.82 per cent emotion; estimated AT after uncomfortable-incongruent = 7.40 ± 0.84 per cent emotion; F1,18 = 8.51, p = 0.001).

Does MAMIP affect response bias in happiness detection?

In a final lme analysis, we asked whether these effects occur at the level of response or stimulus encoding. Following Signal Detection Theory (Macmillan & Creelman, 2004) a pure perceptual effect is supported by the full independence between results on sensitivity and response bias. Specifically, the effect of congruency measured on the perceptual indices of performance (d′ and AT), should result to be absent when measured on the decision index of performance (c). This is what we found, as an lme model on c revealed no significant main or interaction effects: no significant decrement of fit was indeed found when contrasting a full lme model including Congruency, Experiment, and their interaction as fixed effects with a baseline lme model with no fixed effects (rc slightly decreases from 0.83, 95% CI [0.74, 0.89] to 0.76, 95% CI [0.65, 0.84]; χ32=4.8, p = 0.18).

In the present investigation, therefore, there is no evidence that learning and arousal contribute to the perceptual response beyond what emotion intensity and the congruency between action and emotion valence can explain.

Action/emotion valence congruency improves anger detection but does not modify response bias: against valence per se

The results of Experiments 1 and 2 corroborated the idea that the facilitation-by-congruency effect induced by MAMIP was additive, robust, independent of learning and arousal, and perceptually-based: the processing of positive emotional features gets more salient after comfortable (congruent) then comfortable (incongruent) reaches. However, the result cannot be generalized to the entire affect domain, being specific for positive emotions (i.e., happiness), and cannot be interpreted univocally given that, in principle, it might have been produced by action valence per se: with happiness detection being facilitated by a comfortable action sequence, regardless of the correspondence between the valence of the mood induced by the action sequence and the emotion to be detected. If the MAMIP effect is general and independent from action valence then anger detection performance in Experiments 3 and 4 should be facilitated by uncomfortable actions, thus producing an action/emotion congruency effect similar to the one observed in Experiments 1 and 2.

This expectation closely matches results of Experiments 3 (uncomfortable ⇒ comfortable) and 4 (comfortable ⇒ uncomfortable) shown in Fig. 6. The distributions of average d′ resulting from each unique combination of morph intensity (10, 20, 30%), emotion and reaching sessions shown in Figs. 6A and 6E are in strong agreement with an additive (not multiplicative) facilitation-by-congruency effect: in both action ordering conditions (uncomfortable ⇒ comfortable in Fig. 6A; comfortable ⇒ uncomfortable in Fig. 6E) d′ increases linearly as a function of per cent anger in the morph (coded on the x-axis), and is aided by congruency (coded by colors with congruent-uncomfortable in red, and incongruent-comfortable in cyan) by an almost constant amount at increasing per cent anger in the morph values.

Figure 6 Action/emotion valence congruency affects anger detection, but does not modify response bias.

Average d′ triplets (A, E) and average ATs, for the congruent (uncomfortable, in red) and incongruent (comfortable, in cyan) reaching sessions (B, F), together with individual ATs (C, G) and response biases (D, H) changes due to congruency, in Experiment 3 (uncomfortable ⇒ comfortable) and 4 (comfortable ⇒ uncomfortable), as coded by the icons on top. Error bars represent ± SEM. The cyan and red lines in (A, E) are the lme model regression line and the shaded region corresponds to ± standard error of the regression (same interpretation as in Fig. 5). (B, F) A red point closer to zero than the cyan point indicates an absolute anger sensitivity threshold decrement induced by congruency. (C, G) Individual threshold changed between the incongruent and congruent reaching sessions in Experiments 1 and 2, respectively (same axis encoding as in Fig. 5). The vertical green line represents the grand average threshold change due to congruency ± SEM. (D, H) Individual response bias changed between the incongruent and congruent reaching sessions in Experiments 1 and 2, respectively (same axis encoding as in Fig. 5). The vertical green line represents the grand average response bias change due to congruency ± SEM.

This observation was confirmed by an lme analysis testing the effects of Morph Intensity, Action Ordering, and action/emotion Congruency on d′ triplets. The analysis revealed that Morph Intensity (β2 = 0.181 ± 0.0175; F1,93 = 107.25, p = 0.00), and action/emotion Congruency (estimated d′ gain due to congruency = 0.648 ± 0.29; F1,93 = 5.14, p = 0.024) were the only factors affecting anger sensitivity (t = 14.30, df = 112, p = 0.00, r2 = 0.64, 95% CI [0.53, 0.74], rc = 0.77, 95% CI [0.69, 0.83]). No further main effects or interactions were observed (χ52=4.74, p = 0.45). This equal slope lme model best fitting our data set had a negative intercept in the mood congruent condition (β1 = −1.35 ± 0.47; df = 88.25, t = −2.90, p = 0.005, d = 0.64). As shown in panels Figs. 6A and 6E, anger detection sensitivities increased linearly with a similar lme estimated rate in Experiments 3 and 4 of about 1.67 ± 0.24 (Fig. 6A) and 1.98 ± 0.35 (Fig. 6E) respectively, every 10 per cent increment in the neutral-to-angry morph continuum (F2,85 = 0.401, p = 0.67), with an average d′ gain due to congruency that remained constant at increasing morph intensities in a direction consistent with an additive facilitation-by-congruency hypothesis in both experiments (F1,85 = 0.174, p = 0.677): the estimated d′ gain due to congruency measuring about 0.77 ± 0.31 units in Experiment 3 (Fig. 6A, t vs 0 = 2.48, df = 8, p = 0.037, d = 1.76) and 0.58 ± 0.22 units in Experiment 4 (Fig. 6E, t vs 0 = 2.38, df = 9, p = 0.041, d = 1.59).

We further confirmed the direct increase of d′ as a function of morph intensity through congruent and incongruent conditions by post-hoc paired one tailed t-tests. As summarized in Table 3, in Experiment 3, as anger increased from 10 to 20% d′ significantly increased by about 0.83 ± 0.44 d′ units and 1.47 ± 0.65 d′ units, after comfortable-incongruent and uncomfortable-incongruent reaches respectively; similar d′ increments due to a 20 to 30% anger in the morph growth were observed after comfortable-incongruent (t vs 0.83 ± 0.44 = 1.87, df = 9, p = 0.093, d = 1.24) and uncomfortable-congruent reaches (t vs 1.47 ± 0.65 = 1.10, df = 9, p = 0.30, d = 0.73). Post-hoc tests on d′ values from Experiment 4 were strikingly similar, confirming that, independent of Action Ordering, Congruency did not affect the rate of increase of d′ over morph intensity, which is consistent with a rather general additive (not multiplicative) facilitation-by-congruency effect. As by in Experiment 3, also in Experiment 4 any 10% increment in the morph intensity (i.e., from 10 to 20%, or from 20 to 30%) produced a significant d′ gain (see Table 3). Furthermore, a similar d′ gain was induced for 10 to 20% and a 20 to 30% anger increments in the morph by both the congruent-uncomfortable (t = 1.81, df = 8, p = 0.11, d = 1.28) and the incongruent-comfortable (t = 0.79, df = 8, p = 0.45, d = 0.55) sequence of reaches.

Table 3 Summary of post hoc analyses for Experiments 3 and 4.

		Per cent anger range	Mean d′ gain and SEM	Paired t test	
Experiment 3	Incongruent	[10%–20%]	0.83 ± 0.44	t = 2.29, df = 9, p = 0.048, d = 1.52	
[20%–30%]	2.18 ± 0.78	t = 3.21, df = 9, p = 0.011, d = 2.14	
Congruent	[10%–20%]	1.47 ± 0.65	t = 2.69, df = 9, p = 0.025, d = 1.79	
[20%–30%]	2.17 ± 0.86	t = 2.93, df = 9, p = 0.017, d = 1.95	
Experiment 4	Incongruent	[10%–20%]	1.55 ± 0.75	t = 2.53, df = 8, p = 0.035, d = 1.79	
[20%–30%]	1.71 ± 0.80	t = 2.59, df = 8, p = 0.032, d = 1.83	
Congruent	[10%–20%]	1.40 ± 0.68	t = 2.49, df = 8, p = 0.037, d = 1.76	
[20%–30%]	3.25 ± 0.79	t = 4.63, df = 8, p = 0.002, d = 3.27	

The additive facilitation produced by MAMIP on anger detection performance in both action ordering conditions produced a pattern of global sensitivity enhancement and absolute threshold decrements due to congruency that closely resembles the one observed in Experiments 1 and 2. Global d′ after the uncomfortable (congruent) reaching session outperformed those after the incongruent-comfortable reaching session by: 0.24 d′ units, 95% CI [−0.02, 0.50] (estimated global d′ = 0.55 and 0.79, after comfortable and uncomfortable reaching sessions, respectively), in Experiment 3 (F1,9 = 4.18, p = 0.05, d = 1.3), and 0.35 d′ units, 95% CI [−0.01, 0.71] (estimated global d′ = 0.78 and 1.13, after comfortable and uncomfortable reaching sessions, respectively), in Experiment 4 (F1,9 = 8.10, p = 0.02, d = 1.8). These increments in sensitivity due to congruency were statistically similar across action ordering conditions as confirmed by an lme model including the Experiment as fixed effect (t = 13.82, df = 36, p = 0.00, r2 = 0.84, 95% CI [0.71, 0.91], rc = 0.88, 95% CI [0.81, 0.93]). The model indeed revealed a significant main effect of facilitation by congruency of about 0.29 ± 0.09 d′ units (F1,17 = 9.3, p = 0.007), but not of the Experiment (F1,17 = 1.90, p = 0.19), and not of the Congruency × Experiment interaction (F1,17 = 0.35, p = 0.56).

A similar result was obtained on individual per cent anger value above which a difference between the original neutral faces and the faces morphed with anger gets just noticeable; i.e., AT (shown in Figs. 6B and 6F for Experiments 3 and 4, respectively). Again, the lme analysis (t = 6.63, df = 36, p = 0.00, r2 = 0.55, 95% CI [0.31, 0.63], rc = 0.61, 95% CI [0.45, 0.74]), revealed that the only significant factor affecting ATs across experiments was Congruency (F1,17 = 12.35, p = 0.003), which produced an amount of AT decrement that was similar, though significant, in both Experiment 3 (7.1, 95% CI [1.15, 13.03] per cent anger in the morph; F1,9 = 7.27, p = 0.02, d = 1.7, Fig. 6C) and Experiment 4 (7.75, 95% CI [0.00, 15.50] per cent anger in the morph; F1,8 = 5.6, p = 0.03, d = 1.57, Fig. 6G).

Does MAMIP affect response bias in anger detection?

A final lme analysis on response bias c demonstrated that the above discussed effects of MAMIP on anger detection performance were purely perceptual as those observed on the happiness detection task (in Experiments 1 and 2). Again, no significant decrement of fit was indeed found when contrasting a full lme model including Congruency, Experiment and their interaction as fixed effects with a baseline lme model with no fixed effects at all (rc slightly decreases from 0.97, 95% CI [0.94, 0.98] to 0.96, 95% CI [0.92, 0.97]; χ32=7.0, p = 0.08). We confirmed this by post-hoc analyses showing that the response bias change due to congruency was statistically unreliable in both Experiment 3 (−0.16, 95% CI [−0.32, 0.01], t = −2.01, df = 9, p = 0.075; Fig. 6D) and 4 (−0.038, 95% CI [−0.22, 0.14], t = −0.48, df = 8, p = 0.64; Fig. 6H).

Symmetry of the action/emotion congruency effect

Our experiments univocally demonstrate that responses in our facial emotion detection task are, perceptually (not cognitively) determined by displayed emotion intensity, as well as by the congruency between bodily motor acts and emotion valence. The facilitation produced by emotional congruency, in addition of being independent of the arousing effect of goal-directed reaches (as demonstrated in Experiments 1 and 2), generalizes to the negative domain of affects (as demonstrated in Experiments 3 and 4) being thus also independent of action valence per se. Emotional congruency indeed facilitates the performance when the detection task is preceded by both a positive-comfortable action sequence (coupled with a positive emotion to be detected as in Experiments 1 and 2), and a negative-uncomfortable action sequence (coupled with a negative emotion to be detected as in Experiments 3 and 4).

However, a major difference between our two sets of experiments is revealed by a strong emotion detection anisotropy consistent with an overall lower global sensitivity (average global d′ of 1.49, 95% CI [1.33, 1.64] vs. 0.80, 95% CI [0.63, 0.97] in Experiments 1 & 2 vs. 3 & 4, respectively; Welch two sample t = 6.00, df = 75.11, p = 0.00, d = 1.36). This caused also a higher, though not significantly, threshold (average AT was 6.32, 95% CI [5.08, 7.56] vs. 8.49, 95% CI [5.85, 11.12] in Experiments 1–2 vs. 3–4, respectively; Welch two sample t = −1.51, df = 52.841, p = 0.13, d = 0.34) in the anger rather than in the happiness detection task. Such a bias was in line with results of the preliminary baseline experiment and with growing evidence in the emotion perception literature showing that realistic faces, as those used in the current experiments, often give rise to a happiness (rather than the more often found anger) detection advantage relative to both angry (Becker et al., 2011; Becker et al., 2012; Juth et al., 2005) and sad (Srivastava & Srinivasan, 2010) faces.

In order to reconcile our data across emotions with different valence and to test (consistently with G3) the degree of symmetry of the facilitation-by-congruency effect, we recoded the four lme relationships describing the covariation of d′ as a function of Morph Intensity and action/emotion valence Congruency in our dataset in the generic emotion detection Cartesian space introduced in Fig. 2 (with d′ for happiness detection on the y axis and d′ for anger detection on the x axis). Consistently with an additive facilitation-by-congruency effect, the best fitting lme models resulting from the average d′ by morph intensity relationships observed in Experiments 1 & 2 and Experiment 2 & 3 respectively, were both additively modulated by morph intensity and by congruency independently from action ordering. This gave rise to the following set of four lme regression lines: a couple of regression lines for happiness detection with a common slope β2 happy of about 0.27 and intercepts equal to −1.49, for the action/emotion comfortable congruent condition (β1 happy comfortable), and −2.02, for the action/emotion uncomfortable incongruent condition (β1 happy uncomfortable);

a couple of regression lines for anger detection with a common slope β2 anger of about 0.181 and intercepts equal to −1.35, for the action/emotion uncomfortable congruent condition (β1 anger uncomfortable), and −2.00, for the action/emotion comfortable incongruent condition (β1 anger comfortable);

Entering the parameters of these four sets of lines into Eq. 1 (pairing them appropriately following the procedure discussed in the Rationale and expectation section) allows recoding them into the emotion detection space. As shown in Fig. 7 this procedures defines two parallel lines with slope β2 G = 1.47: the blue solid line, standing for the average performance after comfortable reaches with β1 G = 1.465, and the violet solid line, standing for the average performance after uncomfortable reaches with β1 G = −0.025. Importantly, the position of the grey dot representing the average d′ in the baseline inaction condition relative to line of comfortable and uncomfortable reaches in the emotion detection space revealed an almost symmetric facilitation by congruency effect induced by MAMIP. The average y coordinate of the baseline inaction point (1.26 ± 0.159), indeed, did not deviate significantly, along the happiness dimension, from the corresponding coordinate of the point along the line bisecting the space in between the comfortable and uncomfortable lines (0.935), standing for a perfectly symmetric effect (Welch two sample t = 1.4543, df = 172, p = 0.1477, d = 0.22), while it was: (1) significantly smaller than the corresponding coordinate of the point along the comfortable line (1.68), standing for an asymmetric effect fully in charge of the uncomfortable reaches (Welch two sample t = 1.98, df = 172, p = 0.049, d = 0.32), and significantly larger than the corresponding coordinates of the point along the uncomfortable line (0.19), standing for an asymmetric effect fully in charge of the comfortable reaches (Welch two sample t = 4.75, df = 172, p < 0.001, d = 0.72).

Figure 7 Comfortable and uncomfortable reaches impact sensitivity in an approximately symmetric way relative to a baseline inaction condition.

The average d′ ± SEM in the baseline inaction condition (grey dot), is plotted in the emotion detection space together with average d′ ± SEM obtained in Experiments 1–4 (collapsed across Action Ordering conditions), with average performance to the happiness detection task on the y axis, and average performance to the anger detection task on the x axis. The two oblique solid lines represent the lme estimated d′ calculated on the basis of individual d′ after a sequence of comfortable (blue solid line) and uncomfortable reaches (violet solid line), according to Equation 1; with the blue line (line of comfortable reaches) representing estimated happy congruent (comfortable) d′ as a function of estimated angry incongruent (comfortable) d′; and the violet line (line of uncomfortable reaches) representing estimated happy incongruent (uncomfortable) d′ as a function of estimated angry congruent (uncomfortable) d′. The grey dotted line represents a reference over which the grey baseline dot is expected to lie on if the facilitation-by-congruency effect of action-induced mood on emotion sensitivity is perfectly.

The average coordinate of the baseline inaction point along the anger dimension (0.15 ± 0.049) instead deviates from the corresponding coordinates of all three lines: the line of perfect symmetry (Welch two sample t vs. 0.37 ± 0.049 = 3.20, df = 170, p = 0.002, d = 0.50), the line of comfortable reaches (Welch two sample t vs. − 0.14 ± 0.049 = 4.05, df = 170, p < 0.001, d = 0.62), and the line of uncomfortable reaches (Welch two sample t vs. 0.87 ± 0.049 = 10.45, df = 170, p < 0.001, d = 1.60).

Present results demonstrate that both comfortable and ucomfortable actions impact the perception of facial expression. However, while the facilitation-by-congruency effect on happiness detection is balanced across different types of actions, anger detection is facilitated by action induced mood congruency in a slight unbalanced way, with uncomfortable reaches being slightly more effective than comfortable reaches.

Conclusions

The present study demonstrates that the internal state of comfort/discomfort induced by reaching affects the detection of facial expressions in a direction consistent with the congruency between the valence of the action induced transient mood and the target emotion. Performance in a facial emotion detection task was indeed facilitated by congruent couplings between the valence of bodily actions performed before the task and emotions. This was revealed by a sensitivity enhancement and a consistent threshold decrement for facial expressions of emotion congruent with the valence of bodily motor acts, despite the absence of significant shifts in response bias: with happiness detection being facilitated by a sequence of comfortable reaches (Experiments 1 and 2), and vice versa with anger detection being instead facilitated by a sequence of uncomfortable reaches (Experiments 3 and 4). Importantly, these effects were consistent with an additive (not a multiplicative) facilitation-by-congruency effect, being the performance increment due to congruency almost constant at increasing morph intensities for both positive and negative target emotions. Notably, neither arousal by motor activation (predicting the opposite results in Experiments 1–2) nor action valence (comfort/discomfort of bodily action per se) can account for such effects. Furthermore, the systematic sensitivity changes produced by MAMIP did not cause analogous changes in response bias, demonstrating a full dissociation in our task between the way the internal states induced by action affect stimulus encoding (i.e., perception) vs. response selection (i.e., decision).

We interpreted such a dissociation as a compelling evidence for a true top-down effect (in particular, action induced transient mood) on perception of tertiary qualities (in particular, perceived facial expressions): facial expression of emotion is penetrated by observer’s internal states induced by bodily action, producing a facilitation by action/emotion valence congruency effect dependent on stimulus encoding rather than on response selection (Spruyt et al., 2002). This is consistent with an encoding stimulus account of affective priming indirectly produced by our bodily actions (i.e., indirect affective priming, see Janiszewski & Wyer, 2014). The reaching sequence performed before the detection task acted as an affectively polarized prime pre-activating the memory representations of affectively related facial features, thus making it easier to encode emotional facial features belonging to the same valence domain rather than to a different one.

Notably, the way comfortable and uncomfortable reaches impacts the perception of facial expression in our study is almost symmetric relative to the baseline inaction condition. This is consistent with the idea that, relative to the neutral mood experienced during inaction, the internal mood state of the actor is effectively modulated by both the musculoskeletal discomfort induced by reaching beyond a critical distance (where the arm is no longer sufficient) (Conway, 1999; Mark et al., 1997) and the empowerment of motor skillfulness induced by reaching within the peripersonal space. This sets the stage for the occurrence of a mood-congruency effect in emotion perception.

Despite the consistency of our results, our study revealed a rather strong happiness detection advantage. This result casts our data on sensitivity-by-affect spaces with different metrics and thus statistically incomparable across emotions. In order to extract a general index (i.e., irrespective of emotion type) of the facilitation-by-congruency effect produced by MAMIP we thus mapped performance in the Cartesian space of action/emotion valence congruency (Fig. 8). In such a space, each individual d′ value relative to detection of an emotion congruent with reaching valence (on the y axis) is coupled with the corresponding d′ value relative to detection of an emotion incongruent with reaching valence (on the x axis). Representing each individual performance after a reaching session (either congruent or incongruent with the target emotion) as three points (one for each morph intensity) in the Cartesian space with the d′ after the congruent condition in the y axis and the d′ after the incongruent condition in the x axis thus provides a compact way to represent performance.

Figure 8 Joining emotion detection performances in the Cartesian space of emotion/action congruency.

Average d′ values resulting from each unique combination of morph intensity, target emotion, and action ordering after the congruent condition (on the y axis) as a function of corresponding average d′ values after the incongruent condition (on the x-axis). Vertical and horizontal error bars represent ± SEM after congruent and incongruent reaches, respectively. A point lying in the green half of such a Cartesian space represents a detection performance facilitated by congruency. The black dashed line cutting the Cartesian space in two equal halves is the reference for evaluating the overall additivity of the effect revealed by all four experiments: only points arranged along a line parallel to such a reference indeed denote an additive effect. In the legend: action ordering is coded by color (cyan for uncomfortable ⇒ comfortable; red for comfortable ⇒ uncomfortable); morph intensity by element size (small, medium, large corresponding to 10, 20, 30% morph respectively); the two target emotions by shape (circles for anger; triangles for happiness). The red, cyan and grey lines are the lme model regression lines (cyan fitting the uncomfortable ⇒ comfortable performances; red fitting the comfortable ⇒ uncomfortable performances; grey fitting the entire data set regardless of the action ordering) and the shaded region corresponds to ± standard error of the regression. The best fitting lme includes the d′ after incongruent reaches as the only predictor and is characterized by a lme regression line (the grey line) with unitary slope and positive 0.345 intercept evidenced at the margin of the graph.

The representation in Fig. 8 is indeed independent from absolute target emotion intensities and thus optimal for testing the facilitation-by-congruency effect in its general form. In the Cartesian space of emotion/action congruency an additive facilitation-by-congruency effect of MAMIP would be characterized by a family of lines with unitary slope and positive intercept: the value of the intercept being proportional to the sensitivity gain produced by action/emotion valence congruency (i.e., d′ after congruent − d′ after incongruent), independent of the emotion to be detected. A multiplicative facilitation-by-congruency effect would instead be characterized by a family of lines with null intercept and positive slope: the value of the slope being proportional to the relative gain produced by action/emotion valence congruency (i.e., ratio between d′ after congruent and d′ after incongruent).

The co-variation found in our data between d′ after congruent and incongruent reaches across types of target emotion, morph intensities and actions ordering conditions was clearly consistent with a model of the MAMIP effectiveness based on an additive (not multiplicative) facilitation-by-congruency hypothesis. This is shown in Fig. 8, where we recoded our 24 average d′ into 12 points distributed linearly along the Cartesian space of emotion/action congruency. This allowed us to predict individual detection performance after the action/emotion valence congruent condition (y axis) by means of the individual detection performance after the action/emotion valence incongruent condition (x axis) through lme regression. In both action ordering conditions, the 6 pairs of points (6 circles for uncomfortable ⇒ comfortable; 6 triangles for comfortable ⇒ uncomfortable) representing the average joined performance with positive and negative emotions are all placed in the positive half of the Cartesian space of emotion/action congruency and are well aligned along regression lines, fitting individual performances (N = 117) with unitary slope (uncomfortable ⇒ comfortable, cyan line: 1.0135 ± 0.0237; t = 0.760, df = 83.83, p = 0.45; comfortable ⇒ uncomfortable, red line: 0.96 ± 0.018; t = −1.90, df = 83.86, p = 0.07) and similar positive intercept (uncomfortable ⇒ comfortable, cyan line: 0.35 ± 0.11; t = 3.70, df = 83.83, p = 0.00; comfortable ⇒ uncomfortable, red line: 0.34 ± 0.10; t = 4.306, df = 83.86, p = 0.00).

This is confirmed by the results of the lme analysis testing the effects of the type of emotion, action orderings and morph intensity on individual d′ after congruent reaches, once the effect of individual d′ after incongruent reaches is controlled. (t = 85.41, df = 102, p < 0.001; r2 = 0.986, 95% CI [0.978, 0.99], rc = 0.99, 95% CI [0.989, 0.995]). In this model, the likelihood of sensitivities to the detection of (general) emotion after congruent reaches was explained only by sensitivities to the detection of (general) emotion after incongruent reaches (β = 0.74 ± 0.15, F1, 82.85 = 88.511, p < 0.001). Furthermore, no significance decrement of fit was found when contrasting this model with a model (Fig. 8, grey regression line) with d′ after incongruent reaches as the only covariate (t = 86.1, df = 102, p < 0.001; both r2 and rc remained unvaried to 0.986, 0.99, 95% CI [0.989, 0.995] respectively; χ22=0.34, p = 0.84). This model resulted to have both a unitary slope (0.986 ± 0.015; t = −0.702, df = 167.9, p = 0.484) and a positive intercept of about 0.345 ± 0.075 generic d′ units (t = 5.308, df = 167.6, p < 0.001).

In our study, therefore, there is no evidence that morph intensity, actions ordering and type of emotion per se contributes to the determination of d′ after congruent reaches beyond what d′ after incongruent reaches can explain. Furthermore results were strikingly consistent with the predictions rising from a general additive facilitation-by-congruency effect of MAMIP. According to our Cartesian space of emotion/action congruency indeed: (1) the unitary lme estimated slope demonstrates the additive effect of MAMIP producing constant d′ increments in congruent over incongruent conditions at increasing per cent emotion in the morph (regardless of target emotion); (2) the positive intercept, being equal to 0.345, denotes the constant gain in the facial emotion detection performance produced by a congruency between the valence of the action preceding the detection task and the target emotion.

Notably, the 0.345 Congruency Constant (0.345-CC) is expressed in a generic d′ scale independent of the valence of the emotion to be detected, being it representative of the action induced congruency advantage for both positive and negative facial expression of emotions. The 0.345-CC thus quantifies the generic gain induced by bodily comfort over discomfort associated to motor actions on the detection of subtle variations in facial expressions of emotions with positive over negative valence (and vice versa), occurring in the absence of significant shifts in response bias. The 0.345-CC is thus conceivable as the first general constant of how bodily actions regulate affective perception.

Our additive facilitation-by-congruency effect thus has several close relatives with accounts suggesting that different kinds of memory processes (from implicit to explicit) provide a strong linkage between the perceptual representation of a scene, the action plan representation, and the motor simulation (Barsalou, 2003; Nummenmaa et al., 2014). These accounts are indeed all related to the body of literature suggesting that the visual perception of objects and/or context can prime compatible/congruent actions/representations. As a case point, Tucker & Ellis (1998) found faster left- and right-hand responses when the agent was asked to make a decision about an object that could be grasped with the left and right hands, respectively (but see also, Tipper, Howard & Jackson, 1997; Glover et al., 2004). Similarly motor planning and execution has been found to be affected by different other aspects of the context like social intention (Becchio et al., 2008a; Becchio et al., 2008b; Sartori et al., 2009; Ferri et al., 2011; Quesque et al., 2013), social status), end-goal accuracy (Ansuini et al., 2006), and motor affordances determined on the basis of biomechanical compatibility, relative to size, shape, and material properties of the object-hand system (Mon-Williams & Bingham, 2011; Flatters et al., 2012; Holt et al., 2013). In all these studies, aspects of the context make more accessible the memory specific features of actions that according to the influential planning–control model of actions (Glover, 2004) inform the planning component of prehensile movement. Following on the present results, we conjecture that the processing of tertiary qualities might be informed by a mechanism similar to the one activated by the viewing of a “right-hand” feature that primes a congruent right-handed motor response.

Our findings shed light on the current debate voiced by Firestone & Scholl (2015), between a more traditional “modular” view of perception, according to which visual processing is encapsulated from higher-level cognition (Fodor, 1983) vs. a tidal view of perception, according to which visual processing do instead access to more information elsewhere in the mind than has traditionally been imagined (Goldstone & Barsalou, 1998). Specifically, our evidence in favor of a (limited) penetrability of perception challenges the bold claim that “cognition does not affect perception” (Firestone & Scholl, 2015; main title), and supports the idea that observer’s states linked to the valence of performed bodily acts might act as indirect affective primes, modulating stimulus encoding rather than response selection (Spruyt et al., 2002): bodily actions might prime contents that have favorable or unfavorable motor implications and activate general evaluative concepts (e.g., positive vs. negative), thus affecting object properties experienced as external (i.e., perceptual in the phenomenological sense) and yet loaded with meaning.

Our effect is theoretically relevant for the field of perceptual and cognitive sciences, although the existence of effects of observer’s states on tertiary qualities should not look revolutionary (Firestone & Scholl, 2015; section 4.2). Tertiary qualities–as defined in the Gestalt literature (Köhler, 1938; Metzger, 1941; Sinico, 2015; Toccafondi, 2009)–normally imply a reference to the observer, as reflected in the naïve psychology idea captured by “Beauty is in the eye of the beholder.” Nevertheless, they are phenomenally objective (i.e., perceived as belonging to the object; Köhler, 1929) and show a remarkable–though not exclusive–dependence on configural stimulus properties. Therefore, assessing the extent and direction of observer-dependent effects on tertiary (in particular, expressive) qualities represents an important contribution to perceptual science, which can/must tolerate–we believe–some circumscribed leakage of cognition into perceptual apartments, consistent with grounded cognition (Barsalou, 2010; Kiefer & Barsalou, 2013) among other perspectives.

Finally, the present study complements the study by Fantoni & Gerbino (2014) and provides further evidence that mood congruency mediates the effects of motor action on perceived facial emotions, further showing the potential of MAMIP as an innovative and effective tool for the investigation of embodied cognition.

Supplemental Information

Supplemental Information 1 Data from the four experiments.

Three worksheets are included in the file: (1) RAW_DATASET, with the set of 64 yes/no responses characterizing each individual raw performance; (2) glm values, with individual d′ triplets associated to the three combinations of (N) and (S +N) trials (0–10, 0–20, 0–30% emotion in the morph), for congruent and incongruent conditions; (3) est.Grouped indices, with individual global d′, AT, and c values.

Click here for additional data file.

We thank Matteo Manzini for helping with data collection.

Additional Information and Declarations

Competing Interests

Author Contributions

Ethics

Data Deposition

The authors declare that they have no competing interests.

Carlo Fantoni conceived and designed the experiments, performed the experiments, analyzed the data, contributed reagents/materials/analysis tools, wrote the paper, prepared figures and/or tables, reviewed drafts of the paper.

Sara Rigutti performed the experiments, contributed reagents/materials/analysis tools, reviewed drafts of the paper.

Walter Gerbino conceived and designed the experiments, analyzed the data, reviewed drafts of the paper.

The following information was supplied relating to ethical approvals (i.e., approving body and any reference numbers):

The study was approved by the Research Ethics Committee of the University of Trieste (approval number 52) in compliance with national legislation, the Ethical Code of the Italian Association of Psychology, and the Code of Ethical Principles for Medical Research Involving Human Subjects of the World Medical Association (Declaration of Helsinki). Participants provided their written informed consent prior to inclusion in the study. The Ethics Committee of the University of Trieste approved the participation of regularly enrolled students to data collection sessions connected to this specific study, as well as the informed consent form that participants were required to sign.

The following information was supplied regarding data availability:

The raw data has been provided as Supplemental Material.

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
