# Peer review of "Bodily action penetrates affective perception"

_PeerJ, doi:10.7717/peerj.1677_

## Round 0.1 · original submission · Major Revisions

· Academic Editor

Major Revisions

Dear Dr. Fantoni,

I write regarding your manuscript "Bodily action penetrates affective perception", submitted to PeerJ. Attached below are the comments from two reviewers I asked to evaluate your work. As is my custom, I also read your manuscript carefully before and after reading the reviews.

The two reviewers agree that the topic is interesting and that the experiments are well designed. The reviewers differed, however, in the quality and quantity of the concerns they raised. Although Reviewer 2 is optimistic about the publishability of this work (“minor revision”), Reviewers 1 expressed more doubts and made a "major revision" recommendation.

Based on the reviewers' recommendations and on my own reading, I am rejecting the current version of the paper but I am giving you the option to revise the manuscript for resubmission. I plan to ask the current reviewers to evaluate the revision, if they are available.

Because the two reviews are thoughtful and quite clear, I refer you to these reviews for specific comments. In the rest of my action letter, let me highlight some issues that I consider crucial for your revision.

1) presentation of the results. The results section is very complex and difficult to follow. Results should be presented more clearly, benefiting more of the help of tables and figures. Both reviewers provide helpful suggestions that you can follow to improve the results section and render it more easy to follow.

2) production phase of the experiments. More information should be given on the production phase of the experiments, and the criteria followed to distinguish confortable from unconfortable movements should be clarified. Furthermore, you should clarify why you choose to focus your analyses on accuracy and not on RTs (see comments of Reviewer 1). Finally, you should either include a control neutral condition or at a minimum explain why you decided to avoid including it.

3) screening of participants. More information on how you screened participants should be introduced (see comments of Reviewer 2).

4) discussion. In the discussion you refer to studies showing that visual stimuli activate motor responses (affordances). Given than you found an effect of action on perception, you might want to refer more extensively to recent literature showing that object affordances are modulated by the context.

Please note that, although the reviewers raised many issues, I do not insist that you implement all of their suggestions. I ask, however, that you carefully consider each point and implement the ones that you believe would help strengthen the manuscript. Of course, if you decide against implementing some suggestions, please explain the rationale in your cover letter.

Thank you for submitting your interesting work to PeerJ. I am looking forward to your revised paper.

AnnaM. Borghi, Associate Editor

Reviewer 1 ·

Basic reporting

The text is not always clear and unambiguous. While within the Result section this might be due to the presence of numbers and text intermixed, within the Introduction and Discussion section this lack of clarity might be due to the presence of obscure sentences. Please refer to my general comments for specs on this matter. Related to this, the authors refer quite often to a previous study they performed in 2014. Rather than reporting info about procedure/methods here, the authors refer to their previous publication. Although I understand this might help in packing information thus limiting space, it makes quite hard to follow the text and appreciate what has been done here.

Experimental design

For what concerns methods, it is not clear how the authors control the execution of comfortable/uncomfortable reaches. The paper focuses on methods and results for the discrimination task, but no information is provided about the execution part of the experiment. Related to this, there is no mention of data from reaction times. Finally, I believe that the description of the methods should be improved to have the study fully reproducible by another investigator (e.g., how did the authors control that participants were actually performing the requested movement? Was it self-paced? Did the authors control for trunk’s movement? When did they measure the arm of the participant? Which anatomical reference did they use? Wrist, Shoulder? How was gender distributed within each group? How much time between the reaching and the perception session? And so on).

Validity of the findings

Although my general opinion about the validity of the findings is positive, I am still concerned with three major consideration that I would like the authors to consider when revising the paper. A. In the present study, there is no control, baseline condition, i.e., no reaches before detection task. As a consequence, we do not know whether what found for comfortable movement actually reflects a positive internal state effect or a default mood effect. To me it is hard to believe that every time we move comfortably (I guess the majority of the time), our internal state becomes positive and this lead to a change in how we perceive things. A control – no movement condition could have helped in clarifying this aspect providing a reference level. On the other side the possibility to correlate the degree of comfort (like manipulating the reaching distance in a more continuous and finer way) could have allowed to build a stronger link between comfort of the movement, internal state, and “affective” value of perception. B. the results section is hard to follow. The logic behind the analyses is not always fully clear. There is a lot of numerical information within the section. It is often intermingled with text and this makes almost impossible to follow the text (where does the text begin/end?). My suggestion is to use tables for reporting stat results/average/values, and so on. C. An Additional report on kinematics and reaction time data is needed. In its actual form the paper does not allow to evaluate whether participants actually comply with comfort/discomfort requests (e.g., did they move as to cover the appropriate distance?)Please add information related to kinematics results and RTs.

Additional comments

Within the Introduction, the text is not always fully clear. For instance, at page 2 (lines 103-107) the sentence sounds a bit obscure. Please clarify why the demonstration that kinematics differ depending on their “social” context is relevant for the issue at stake here. In my opinion, the finding that there is a relationship between the internal representation of an action goal and hand kinematics does not imply that hand kinematics (of a comfortable action for instance) would impact on the internal representation of an action goal. Moreover, please provide a reference and a brief description of what the authors mean by “evaluation apprehension” here.

I believe it would be important to provide the reader with some details about the study by Fantoni and Gerbino (2014). In more than one point within the text, the authors refer to methods, set up properties, stimuli used in their previous study and reported on a different paper. Although I understand this is for the sake of brevity, I also believe that this lack of details often result in the difficulty to follow the text and actually understand what has been done here (e.g., why it adds to what has been done before, why current results are in line with what has been found, and so on). At the end of the Introduction, the authors do mention some details about their previous study. My suggestion is to move this description early on in the section and improve its level of detail as to make the reader able to fully appreciate why this study represents a step forward.

Page 3, lines 129 – 139. Please provide the reader with details about what has been found in Kirsch & Kunde 2013 and Volcic et al., 2013. In its current form, this part of the Introduction refers to previous studies investigating the relationship between reaching comfort and perception, but it does not provide info about their main results.

Page 3, lines 140 – 144. Here it is stated that the aim of the study is: “corroborating Fantoni and Gerbino’s results and providing evidence in favor of an encoding rather than response account of affective priming of perceived facial expressions of emotion, as mediated by action-induced mood states”. As I mentioned, at this point of the ms the paradigm by Fantoni and Gerbino has not been described yet, this sentence then does not make much sense. Moreover, the reader might not be familiar with the distinction between encoding and response account. Please add a brief description of what it is meant and why the present study might help in answering this question. Finally, the authors may want to add the date of the study (2014?).

Page 3. When referring to short-long distances for comfortable- uncomfortable reaches, there is no mention of how distances were selected. In other words, I would like to know more about the criteria that led the authors choose these distances as inducing either comfort/discomfort. Do they refer to previous data? Did they take advantage of data from a preliminary study in which participants were asked to rate how much comfortable they felt after repetitive reaches? Is there any previous research addressing this aspect and providing data concerned with subjective estimation of mood after comfortable vs. uncomfortable movement execution? Please clarify.

For what concerns the Rationale and Expectations section. Please rephrase G1 trying to make the concept simpler. To me the following sentence does not sound fully clear: “…as a determinant of perceived facial expressions of emotion”.

Page 6. When referring to “sensitivity experiment” please provide a definition within parentheses.

At page 8 it is specified that the type of signal was treated as a between subject factor to avoid carry over effects. The authors may want to explain what they mean by referring to carry over effects in this context and provide references of the study they have in mind here (“to avoid difficulties related to carry over effects intrinsic to experimental designs in which actors/observers perform actions and/or detect emotions…).

For what concerns the “Participants” section, I would like the authors to add some important information. For instance, how many women/men per Experiment? What do they intend by saying “pseudo randomly” assigned to one of the four experiment. Was this a random allocation or not? Did they consider gender as a criterion for group allocation? Please provide age information for each group separately, together with age standard deviation as well as information about how long each experiment lasted.

About the Apparatus, stimuli, & Design section, the authors refer to their previous study (2014) for details about the set-up (e.g., viewing geometry, augmented reality apparatus, and so on). I do understand this is for the sake of brevity, but it is important to have all the relevant information here as to avoid going back and forth from a different paper. Please consider this comment when revising the entire manuscript.

Concerning the reaching execution phase, please improve the description as to clarify: a. which were the instructions given to participants; b. which distance increments were used when manipulating depth range (from 0.65 to 0.75 in step of ??); c. whether or not kinematics data were recorded; d. how the authors control whether participants’ kinematics meet the request of reaching within the comfortable/uncomfortable range? e. If participants could move or not (e.g., I wonder whether trunk’s movement could affect the depth range then altering the perception of comfort).

If I understood correctly, in their study in 2014 the authors take advantage of a control condition in which participants did not move before perceptual task. In the present study this control condition is absent. In my opinion it would have been interesting to see whether the performance would be the same when comparing no movement with comfortable movement condition (and corresponding compatibility with the signal). In other words, while I see that an uncomfortable movement can induce a negative mood, I do not see why a comfortable movement would induce a positive mood. In other words, a comfortable movement is merely a natural movement rather than a “positive” movement. If this would be case, every time we move within the normal/comfortable depth range, our perception of external stimuli would differ with respect to when no movement had occurred. The authors might want to explain why they did not include a baseline-no movement condition in the paradigm. In my opinion, with no control condition, effects from happy/comfortable congruency circumstances should be reframed.

For what concerns the “Procedure” section, the authors refer to “successful reaches”. However, it is not clear according to which criterion a reaching movement would be considered either successful or not successful. As a general comment, the description of movement recording should be improved. At the moment it is stated kinematics data were acquired but there is no mention of kinematics results within the manuscript. Moreover, I wonder why the authors did not present participants with the same amount of “signal present” and “signal absent” trials. It seems they used a 3:1 ratio for signal administration. Later on it is stated that participants were explicitly told about this asymmetry. In my opinion, this might bias their behavior, inducing the adoption of a strategy (like a counting strategy for instance) to accomplish the task. Can the authors explain why they opted for this “asymmetrical” solution?

About the structure of the results section, in its current form the section is not always easy to follow. My main concern refers to how each analysis and corresponding results are presented. The section is very long, the rationale is not always linear because from time to time the authors introduce tests/explanations of why results differ between one experiment and the other (e.g., as for different sensitivity for happiness and anger or for action ordering in exp 1 and 2). It seems that the flow is dictated by what they found in every analysis rather than from the overall prediction design they had in mind when running the experiment. However the number of results, questions to answers, stat models and graphs is quite high so that it is crucial to help the reader in keeping in mind why a given test has been done, which answer it provides, how, and so on. For instance, there are two sections; namely “Evidence from the raw distribution of yes responses” and “Quantitative analysis”. First, it is not clear what the authors mean by “raw data”. I was expecting for data by each subject and no mention of statistical analyses within this section. Rather mean data and ANOVAs results are actually reported here. Beside this, the presentation of the results can be made more straightforward (e.g., using tables for reporting Fs, p values, CI, and so on). One possibility is to present the data in different sections (or sub sections), one section for each experimental question (e.g., 1. Does action/emotion congruency impact on response precision? 2. Is this an additive or a multiplicative effect? 3. Is this a perceptual or a post-perceptual effect?).

If I look at Figure 2, it seems that when no emotion was there (0%) participants tend to say “yes” more when the emotion to detect was incongruent than when it was congruent (cyan line higher than red one in all four panels). Am I correct? Was there a significant effect? If so, why would this be the case. Related to this, it seems there is a greater variability in data from experiment 1 & 2 than 3 & 4. Please provide a tentative explanation of this discrepancy (if confirmed).

Lines 647-655. Why the effects should be mediated by the level of experience with the detection task and the face stimulus set? Perhaps I misunderstood, but in my opinion this consideration opens to the possibility that experience might lead to different results thus suggesting that what reported here might not be fully “stable”. Related to this, I noticed that in the training session, participants were not exposed to the same trials as in the experimental session (that is: 0 and 50% of emotion rather than the four “experimental” intervals). Can the authors explain why they did not expose the participants to the very same stimuli as in experimental session? At the end of the training session participants had seen 0% emotion faces but not the other emotion levels. To me this might signify that they were more familiar with 0% than with 10%, 20%, and 30% faces. Please explain.

Please improve the description of the rationale behind the usage of performance mapping within the Cartesian space. As it stands it is unclear why the authors perform it. Please focus on explaining what this would add to previous results and why. Right after the description of this mapping the authors get back to the addictive / multiplicative issue (page from line 837). I strongly advise the authors to “pack” the results as to make them more straightforward.

Lines 746. …”after comfortable-incongruent and uncomfortable-incongruent…”. I guess there is a typo. I think there is also something wrong in labeling at lines 759 and 760.

From line 840 to 846. This passage sounds quite complex. As previously stated, I believe all these analyses would benefit from a detailed description of their rationale together with a clear explanation of what they add to what previously described/demonstrated in terms of stat models/results.

Line 919. I am not fully convinced about the following sentence: “….the present study demonstrates that the internal state of comfort/discomfort induced by reaching affects…”. With no control condition (no reaches), with no data indicating that a comfortable reach actually induces a positive internal state, I wonder whether we can state that “the present study demonstrates that the internal state of comfort induced by…” . To me a stronger argument would be necessary. In this respect, it is a pity we do not have any data from a debriefing exit-questionnaire. It would have been interesting to know participants’ opinion about a. goal of the study; b. how they feel after 50 uncomfortable/comfortable. The authors may want to improve the discussion in the attempt to demonstrate that their paradigm/data do allow to conclude that comfortable reaches induce positive internal state and, in turn, an impact on “affective value” of perception.

Lines 913-916. “…thus it quantifies the generic gain induced by bodily comfort over discomfort associated to motor actions on the detection of subtle variations in facial expressions of emotions with positive over negative valence (and vice versa...”. Related to my comment about the lack of a control condition I wonder whether it is possible to state that it is because a “bodily comfort” occurs. As previously mentioned, I am not sure it is possible to assume that a natural reach to grasp contributes to a subjective experience of comfort.

As for kinematics data, there is no mention of reaction time data within the manuscript. It would be interesting to see whether the results for accuracy match those for RTs (e.g., is there any speed-accuracy trade off?).

Lines 966-967. Please rephrase this sentence since it sounds a bit obscure: “…and support indirect affective priming as a candidate mechanism by observer’s states mold object properties experienced as external (i.e., perceptual in the phenomenological sense) and yet loaded with meaning”.

Reviewer 2 ·

Basic reporting

The paper is well written and follows expected conventions of scientific writing. I do have some concern with the rationale for the experiment based on the supporting literature used. At times it is hard to follow the logic.

Experimental design

The design is sound and appropriate for the questions being asked.
I would like the methods section to include:
Where the participants screened for any kind of mood disorder? Did they have any history of mood disorder and was there any screening for the use of antidepressant medications?
What was the rationale for the morphing phases?
The authors state that there was a 90% correct response cut off to enter the experimental phase for emotion detection. Where all participants’ data then used regardless of response rates post training or was there a performance cut of applied again in the experimental phase?
Inclusion of a figure illustrating the emotional expression identification task would be helpful.

Validity of the findings

The findings are valid however the complexity of the analysis and reporting makes it hard to follow the experimental logic at times. A summary table of the results would help to clarify them especially as the analysis reporting is complex and extensive.
Inclusion of a table of statistical results would make them easier to follow.

Additional comments

At times the manuscript moves from being over simplistic in the introduction and conclusions to overly complex in the analysis and results.

---

## Round 0.2 · accepted · Accept

· Academic Editor

Accept

I have received the comments from one of the original reviewers. I agree with the reviewer that the manuscript is greatly improved. Even if still very complex, I think it now represents a very interesting and sophisticated contribution to the field

Reviewer 2 ·

Basic reporting

The article is much improved and meets the criteria for basic reporting. Pass.

Experimental design

Clarification has improved understanding of the design. Pass.

Validity of the findings

Findings, albeit complex and sometimes hard to follow are sound. Pass.

Additional comments

The manuscript is greatly improved. It is easier to read and clarification of the methods and results makes it easier to follow the experimental logic.